# Non-equilibrium quantum transport in presence of a defect: the non-interacting case

**Marko Ljubotina, Spyros Sotiriadis⋆ and Tomaž Prosen**

Department of Physics, Faculty of Mathematics and Physics,
University of Ljubljana, Ljubljana, Slovenia

⋆ sotiriadis@fmf.uni-lj.si

## Abstract

We study quantum transport after an inhomogeneous quantum quench in a free fermion lattice system in the presence of a localised defect. Using a new rigorous analytical approach for the calculation of large time and distance asymptotics of physical observables, we derive the exact profiles of particle density and current. Our analysis shows that the predictions of a semiclassical approach that has been extensively applied in similar problems match exactly with the correct asymptotics, except for possible finite distance corrections close to the defect. We generalise our formulas to an arbitrary non-interacting particle-conserving defect, expressing them in terms of its scattering properties.



# 1  Introduction

The problem of quantum transport is an ubiquitous topic of non-equilibrium statistical physics. Transport phenomena are often discussed in the linear response context, where they essentially reduce via Green-Kubo theory to studies of dynamical correlations in equilibrium. On the other hand, genuine non-equilibrium treatment [1,2] requires a subtle control over the thermodynamic limit or, more precisely, over an infinite number of degrees of freedom composing the so-called thermal (or particle/magnetic) baths (or reservoirs).

In the non-equilibrium protocol, which has become known as the *inhomogeneous quench problem*, one prepares an initial state of an infinite low dimensional system (say, a one-dimensional chain) which is out of equilibrium and is characterised by some spatial profile of local physical observables, such as density, temperature, magnetisation, etc (see [3,4] for recent reviews and references). Usually, for simplicity, one considers an initial step profile protocol in which the initial state is given as a product of two distinct equilibrium states for the left and the right semi-infinite part of the system with respect to some point of origin, however more general (say, smooth) profiles can also be considered. It has been shown that a quantum quench starting from such types of inhomogeneous initial states allows for an exact derivation of transport properties in closed-form expressions, either in conformal field theory framework [5–9], where it results in a Stefan-Boltzmann radiative transfer, or for free fermions [10–19], or even in general integrable systems, where the problem can be effectively treated in terms of the so-called *Generalised Hydrodynamics* [20, 21]. The quantum dynamics after such an inhomogeneous quench has been recently observed in ultra-cold atom experiments [22].

The quantities of central interest for the study of quantum transport in such inhomogeneous out-of-equilibrium settings are the *particle current* and *density* (or current and density of any other quantity that satisfies a local conservation law) at fixed position and large time, as well as their *asymptotic profile* at large times and distances with fixed ratio of distance over time. For *ballistic* transport, which is the typical case for non-interacting or integrable systems, these profiles are characterised by the formation of two (or more, for multiple types of particle excitations) *fronts*, moving ballistically to the left and right direction, and the relaxation to a *Non-Equilibrium Steady State* (NESS) in the middle region between the oppositely moving fronts [23]. A NESS is a statistical ensemble that gives the large time values of local observables at fixed position. More generally, it has been shown that a different ensemble can be associated with the asymptotics at each ray of fixed distance/time ratio [15].

As the inhomogeneous quench protocol intrinsically breaks the translational invariance of the problem, it is natural to ask what are the effects of translational symmetry breaking terms in the Hamiltonian, say point (or localised) *defects* at the contact between the two semi-infinite systems, prepared in two distinct equilibrium states as in the case of the step profile quench protocol. Such defects can be interacting or non-interacting, integrability-preserving (such as in the famous Kondo [24] or Anderson impurity problems [25]) or integrability-breaking [26,27]. The latter could be considered as minimal models of a quantum Kolmogorov-Arnold-Moser theorem scenario. General ideas for the treatment of transport in impurity models have

been put forward in the past. At low temperatures Kane and Fisher [28] studied a Luttinger liquid with a weak link and found a non-trivial phase diagram, where RG flows either to a vanishing or a unit quantised conductance. Several approaches have been developed based on integrability [29–31] (see also [32] for a review and [33,34] for more recent discussions) and more recently using conformal field theory [35]. A bosonisation treatment has also been used to study energy transport after joining two XXZ spin chains with different parameter values [36]. Another approximate approach is based on Landauer's theory [37] (see also [38] and references therein for related numerical works), which refers to quantum transport through an impurity in contact with two thermal baths at different chemical potential. The Landauer-Büttiker formula describes the steady state current through the impurity as a function of the voltage (Fermi level difference). However, explicit results for the unitary dynamics of the *whole* closed system *far from equilibrium* (i.e. for strong inhomogeneity bias) are rare and limited to hard-wall defects [15, 39, 40].

On the other hand, many analytical studies of the inhomogeneous quench problem are based on the so-called *semiclassical* or *hydrodynamic* approach. This approach and its various extensions have been widely used mostly in essentially non-interacting systems, in both homogeneous [41–48] and inhomogeneous settings [13, 17–19, 49], but more recently also in genuinely interacting integrable systems [20, 21, 48, 50–54]. In defect related out-of-equilibrium problems it has been applied to the special cases of a quench in the presence of a hard-wall boundary [15] and of a local quench induced by a moving defect [55]. Even though it is not a priori obvious why this approach should be anything more than an approximation, it is conjectured that it gives correct predictions for the exact asymptotics of physical observables at large times and distances, as it turns out to be in perfect agreement with all available numerics.

In this work we study inhomogeneous quenches in the presence of non-interacting defects, which can be studied in an *exact* and mathematically *rigorous* way. We focus on a one-dimensional lattice system of free hopping fermions (equivalent to the XX model of spin $\frac{1}{2}$) with a non-interacting particle-number preserving defect. Our goal is to derive from first principles the exact asymptotics of the particle current and density (not only close to the defect, but in the whole system), which we do using a new analytical approach that handles rigorously the thermodynamic and large time/distance limit. In this way we test the validity of the semiclassical approach, demonstrating that it reproduces correctly the exact results *away from* the defect, but also identifying in certain cases corrections *close to* the defect that cannot be captured by it. Such corrections are present, for example, when the defect produces localised bound state excitations. These corrections consist in persistent oscillations of physical observables, whose strength decays exponentially with the distance from the defect, and in different time-averaged values in comparison with those away from the defect. As physically expected, the defect obstructs the motion of particles diminishing transport. As a result, we find that the initial density difference on the left and on the right side is partially preserved in the steady state. All of the above can be clearly seen in Fig. 1 showing spacetime plots of the particle density and current for a typical case of such an inhomogeneous quench discussed in detail below. Our analysis shows that the dependence of the current/density profiles on the defect strength can be expressed solely in terms of its *scattering* properties, i.e. the reflection or transmission probability that characterises it.

The paper is organised as follows: In Sec. 2 we present the semiclassical or hydrodynamic approach to quantum quenches and apply it to the present problem finding simple formulas for the asymptotics of particle density and current. We also discuss the analogy with Landauer's theory, demonstrating the compatibility of the semiclassical results with the Landauer-Büttiker formula in the special case when the initial momentum densities on the left and right side are thermal. In Sec. 3 we proceed to the exact analytical calculation of the asymptotics of current and density profiles. In Sec. 4 we compare our results with numerics finding perfect

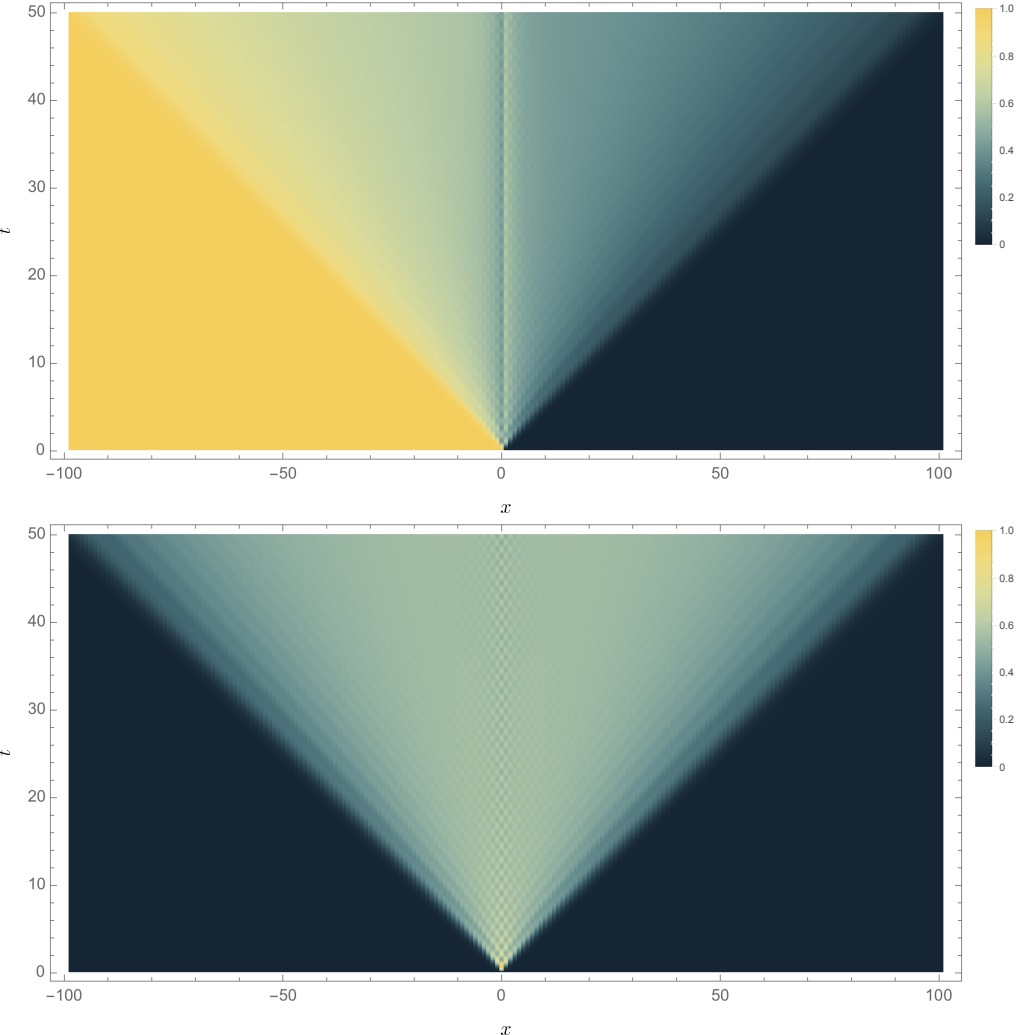

Figure 1: Particle density $n(x,t)$ (*top*) and current $J(x,t)$ (*bottom*) as functions of the position $x$ and time $t$ for the inhomogeneous quench protocol considered in Sec. 3 (numerical data). The initial state has different density on the left and right side and the dynamics is that of a free fermion lattice system in presence of a hopping defect at the origin (numerical data for ratio of defect over bulk hopping $j = 1.2$; values rescaled between 0 and 1; see Fig. 6 for line plots). Notice the ballistic motion of the fronts, the partial preservation of density difference between the two sides of the defect and the finite distance effects close to it (density corrections and persistent oscillations that decay exponentially with distance from the origin).

agreement. In Sec. 5 we generalise the analytical calculation to any non-interacting defect and compare with the semiclassical formulas. We summarise our conclusions in Sec. 6. Effects at finite distance from the defect, partially due to the presence of bound states, are discussed in App. A. Some mathematical tools that are crucial for the rigorous derivation of the results are presented in detail in App. B.1 and B.2.

## 2 Semiclassical or Hydrodynamic approach

The asymptotics of local observables after a quantum quench can be successfully captured by a simple quasiparticle picture, known as semiclassical or hydrodynamic approach. In this section we present a description of this approach and apply it to the present problem of an inhomogeneous quench in the presence of a non-interacting defect, deriving explicit expressions for the current and density asymptotics. We later compare these with the exact results and confirm their validity.

According to the semiclassical approach, at least as far as the asymptotic behaviour of the system is concerned, its dynamics can be derived by considering a statistical system of classical quasiparticles that are produced at the time of the quench and subsequently travel ballistically throughout the system. The quasiparticles correspond to those that would describe the excitations of the bulk part of the post-quench Hamiltonian and their velocities are equal to the corresponding group velocities of excitations. In the present problem the bulk Hamiltonian is non-interacting i.e. there are no quasiparticle collisions which could modify or "dress" their velocities.

For inhomogeneous quenches, the initial probability distribution $\rho_0$ of the created quasiparticles depends on both their position $x$ and momentum $k$ and is deduced for each position from the local values of the quench parameters. More specifically, for a step-like initial state we have

$$\rho_0(x, k) = \rho_{0L}(k)\Theta(-x) + \rho_{0R}(k)\Theta(+x), \tag{1}$$

where the momentum distributions $\rho_{0L/R}(k)$ denote the densities that correspond to homogeneous quenches with initial states given by the left and right side restriction of the actual inhomogeneous initial state. To be specific we will focus on a special initial state with flat densities

$$\rho_{0L}(k) = \nu - \mu, \qquad \rho_{0R}(k) = \nu + \mu, \tag{2}$$

but the following results are valid in general.

The time evolution of local observables, like the density $n(x, t)$, is then calculated tracing the classical trajectories of the quasiparticles and integrating the number of those reaching the point $x$ at time $t$ in all possible ways. In absence of the defect, this means integrating over all initial points $x'$ from which these quasiparticles may originate. In the presence of the defect on the other hand, we should also take into account the fact that quasiparticles coming from the opposite side have first been transmitted through the defect and some of those coming from the same side can now get reflected off the defect and change direction. These scattering effects amount to introducing weights for the contribution of those groups of particles, equal to the transmission and reflection probabilities $T(k)$ and $R(k)$ respectively. The two cases are schematically represented in Fig. 2.

We first focus on the simpler case where no defect is present and next study the modifications introduced by the presence of the defect.

### 2.1 Absence of defect

According to the above arguments, in the absence of the defect the density is given by

$$n(x, t) = \int_{-\pi}^{\pi} \frac{dk}{2\pi} \int_{-\infty}^{+\infty} dx' \, \rho_0(x', k)\delta(x - x' - v(k)t),$$

where $v(k)$ is the group velocity, which in the non-interacting case is given by the derivative of the dispersion relation $v(k) = dE(k)/dk$. Performing the integral over the initial quasiparticle

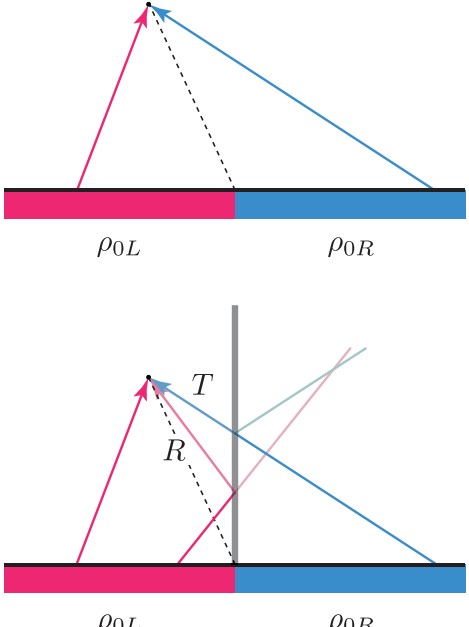

Figure 2: Semiclassical picture for an inhomogeneous quantum quench in the absence (*top*) or presence of a defect at the origin (*bottom*).

position $x'$ and substituting (1) we find

$$n(x,t) = \int_{-\pi}^{\pi} \frac{dk}{2\pi} \rho_0(x - v(k)t, k)$$
$$= \int_{-\pi}^{\pi} \frac{dk}{2\pi} \left[ \rho_{0L}(k)\Theta(-x + v(k)t) + \rho_{0R}(k)\Theta(+x - v(k)t) \right].$$

Already at this step, we observe that the semiclassical picture captures several of the general characteristics of the asymptotics. First, due to the step-like form of the initial state, the density $n(x,t)$ (and similarly all other local observables) is actually a function of a single variable, the "ray" ratio $\xi = x/t$, instead of $x$ and $t$ separately. Second, the value of joint position-momentum density on each of the rays of fixed ratio $\xi$ is given by a $\xi$-dependent momentum distribution, i.e.

$$n(x,t) = \int_{-\pi}^{\pi} \frac{dk}{2\pi} \rho_\infty(k; x/t), \tag{3}$$

with

$$\rho_\infty(k; \xi) = \rho_{0L}(k)\Theta(v(k) - \xi) + \rho_{0R}(k)\Theta(-v(k) + \xi) \tag{4}$$
$$= \begin{cases} \rho_{0R}(k) & \text{if } v(k) < \xi \\ \rho_{0L}(k) & \text{if } v(k) > \xi. \end{cases}$$

In the simplest case of a monotonously increasing function $v(k)$, $\rho_\infty(k; \xi)$ has a simple stepwise form: for $v(k) > \xi$ it is equal to the initial momentum density of the left side of the system, while for $v(k) < \xi$ it is equal to that of the right side (Fig. 3). The threshold momentum $k_*$ separating the two contributions for each ray is therefore the $\xi$-dependent solution of the equation

$$v(k_*) = \xi. \tag{5}$$

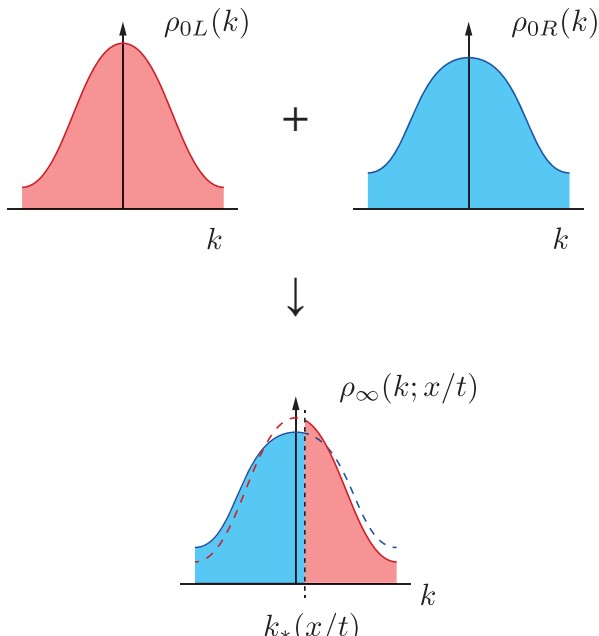

Figure 3: The asymptotic momentum distribution at large distances and times is a stepwise combination of the initial ones on the left and right sides.

At the middle, $\xi = 0$, this reduces to the usual observation that the NESS is built out of right-moving modes carrying the initial density $\rho_{0L}(k)$ of the left side (where they come from) and left-moving modes carrying the initial density $\rho_{0R}(k)$ of the right side.

The current $J(x,t)$ can be calculated in a similar way. Taking into account that in the semiclassical approach it is given by

$$J(x,t) = \int_{-\pi}^{\pi} \frac{\mathrm{d}k}{2\pi} \int_{-\infty}^{+\infty} \mathrm{d}x'\, v(k)\rho_0(x',k)\delta(x-x'-v(k)t),$$

we finally find

$$J(x,t) = \int_{-\pi}^{\pi} \frac{\mathrm{d}k}{2\pi} (\rho_{0R}(k) - \rho_{0L}(k)) v(k)\Theta(x/t - v(k)). \tag{6}$$

Alternatively, we can derive the last result directly from the formula for the density using the continuity equation. Indeed in the scaling limit the continuity equation becomes $\partial_t n(x,t) = -\partial_x J(x,t)$ from which we find $J(x,t) = -\int_{-\infty}^{x} \mathrm{d}s\, \partial_t n(s,t)$ and due to the scaling form of the density

$$J(\xi) = \int_{-\infty}^{\xi} \mathrm{d}\xi'\, \xi' \frac{\mathrm{d}n(\xi')}{\mathrm{d}\xi'}. \tag{7}$$

Substituting (3) and (4) we obtain (6).

In the present problem of the XX spin $\frac{1}{2}$ model, the group velocity is

$$v(k) = \frac{\mathrm{d}E(k)}{\mathrm{d}k} = 2\sin k. \tag{8}$$

This is not a monotonous function in the interval $[-\pi, +\pi]$, however it is monotonous in $[-\pi/2, +\pi/2]$ and has symmetric values outside of the interval. Therefore, as long as the initial densities $\rho_{0L/R}(k)$ are even and symmetric with respect to $\pm\pi/2$, we can restrict the momentum integral in (3) between $-\pi/2$ and $+\pi/2$ and write

$$n(x,t) = 2\int_{-\pi/2}^{\pi/2} \frac{\mathrm{d}k}{2\pi} [\rho_{0L}(k)\Theta(-x + v(k)t) + \rho_{0R}(k)\Theta(+x - v(k)t)].$$

The solution of (5) in this restricted interval for $v(k)$ given by (8), is

$$
k_*(\xi) = \begin{cases} -\pi/2 & \text{for } \xi < -2 \\ \arcsin(\xi/2) & \text{for } -2 \leq \xi \leq 2 \\ +\pi/2 & \text{for } \xi > 2. \end{cases}
$$

Using this solution, we find

$$
n(x,t) = \int_{-\pi/2}^{k_*(x/t)} \frac{\mathrm{d}k}{\pi} \rho_{0R}(k) + \int_{k_*(x/t)}^{+\pi/2} \frac{\mathrm{d}k}{\pi} \rho_{0L}(k).
$$

In the special case of constant initial densities (2) we obtain

$$
\begin{aligned}
n(x,t) &= \rho_{0R}\left(k_*(x/t)/\pi + 1/2\right) + \rho_{0L}\left(1/2 - k_*(x/t)/\pi\right) \\
&= \frac{\rho_{0L} + \rho_{0R}}{2} + (\rho_{0R} - \rho_{0L})k_*(x/t)/\pi \\
&= \nu + (2\mu/\pi)k_*(x/t).
\end{aligned}
$$

## 2.2 Presence of defect

We now focus on the case where a defect is present. In the semiclassical approach the quasi-particles are still the same free particles and have the same initial densities as before, but now they can also scatter with the defect. This means that when calculating the time evolution of the density $n(x,t)$, there are two modifications with respect to the previous case. First, the number of quasiparticles with momentum $k$ that come from the opposite side is reduced by a factor equal to the transmission probability $T(k)$, since they have been transmitted through the defect. And second, there are now quasiparticles originating from the same side that have been reflected off the defect with probability $R(k)$ and changed direction of motion.

We therefore find that the time evolution of the density on the left side is given by

$$
\begin{aligned}
n(x < 0; t) = \int_{-\pi}^{\pi} \frac{\mathrm{d}k}{2\pi} \Bigg[ &\int_{-\infty}^{0} \mathrm{d}x' \rho_0(x',k)\delta(x - x' - v(k)t) \\
&+ \int_{0}^{+\infty} \mathrm{d}x' T(k)\rho_0(x',k)\delta(x - x' - v(k)t) \\
&+ \int_{-\infty}^{0} \mathrm{d}x' R(k)\rho_0(x',-k)\delta(x + x' - v(k)t) \Bigg].
\end{aligned}
$$

In the above we used the fact that transmitted quasiparticles keep moving in the same direction with the same velocity, while reflected ones change direction. Substituting (1) and using the fact that $\rho_{0L/R}(k)$ are even function we obtain

$$
n(x < 0; t) = \int_{-\pi}^{\pi} \frac{\mathrm{d}k}{2\pi} \begin{cases} T(k)\rho_{0R}(k) + R(k)\rho_{0L}(k) & \text{if } v(k) < x/t \\ \rho_{0L}(k) & \text{if } v(k) > x/t. \end{cases} \tag{9}
$$

Working similarly for the right side, we find

$$
n(x > 0; t) = \int_{-\pi}^{\pi} \frac{\mathrm{d}k}{2\pi} \begin{cases} \rho_{0R}(k) & \text{if } v(k) < x/t \\ T(k)\rho_{0L}(k) + R(k)\rho_{0R}(k) & \text{if } v(k) > x/t. \end{cases} \tag{10}
$$

Therefore there is a difference between the asymptotic values on the left and on the right of the defect, which is

$$\Delta n = n(0^+) - n(0^-)$$
$$= \int_{-\pi}^{\pi} \frac{dk}{2\pi} R(k)(\rho_{0R}(k) - \rho_{0L}(k)). \tag{11}$$

We similarly find that the asymptotic profile of the current is given by

$$J(x,t) = -\int_{-\pi}^{\pi} \frac{dk}{2\pi} T(k)(\rho_{0R}(k) - \rho_{0L}(k)) v(k)\Theta(v(k) - x/t), \tag{12}$$

which is valid on both sides, as we can easily verify using the relation $R(k) + T(k) = 1$ and the fact that $\rho_{0L/R}(k)$ and $T(k)$ are even functions while $v(k)$ is odd. In the same way we can check that the above expression is symmetric under $x \to -x$.

The current at the middle is

$$J = -\int_{-\pi}^{\pi} \frac{dk}{2\pi} T(k)(\rho_{0R}(k) - \rho_{0L}(k)) v(k)\Theta(v(k)). \tag{13}$$

In the special case of constant initial densities (2) we find that the current at the middle is

$$J = -2\mu \int_{-\pi}^{\pi} \frac{dk}{2\pi} T(k)v(k)\Theta(v(k)). \tag{14}$$

Note that the current is proportional to the difference of the initial density on the left and right halves and has a direction from the side of higher to that of lower density. Also note that the current vanishes when the defect has infinite strength, blocking completely the passage of particles from one side to the other ($T(k) = 0$) and takes its maximum value when no defect is present ($T(k) = 1$). This is of course what we expect from physical intuition, since the defect tends to obstruct the motion of particles and therefore to diminish the value of the current and to partially preserve the density difference between the left and right side that is inherited from the inhomogeneous initial state.

For the hopping defect which we are going to study in more detail, the reflection and transmission probabilities turn out to be

$$R(k) = \frac{(j - j^{-1})^2}{j^2 + j^{-2} - 2\cos 2k}, \tag{15}$$

$$T(k) = \frac{4\sin^2 k}{j^2 + j^{-2} - 2\cos 2k}, \tag{16}$$

where $j$ is the ratio of the defect hopping over the bulk hopping and measures the defect strength. These expressions can be readily found by solving the quantum mechanical problem of an incident particle travelling with momentum $k$ towards the defect and being reflected and transmitted through it. Notice that these expressions are clearly symmetric under inversion of the defect strength $j \to 1/j$. According to the above semiclassical analysis, this symmetry will be manifest in all results describing the asymptotics of density and current.

## 2.3 Comparison with Landauer's theory

We will now compare the inhomogeneous quench problem with that of an open and typically finite quantum system coupled to two thermal baths at different chemical potential, which was studied by Landauer [56]. This theory describes the current through the system as a function of

the voltage, i.e. chemical potential imbalance, and is relevant for the physics of quantum dots and quantum wire junctions. This problem is different from the quench in terms of physical settings and questions: In the quench problem the system is extended and closed (that is, the dynamics is unitary) and we are interested in the spatial profile of observables which are time-dependent for all times. In Landauer's problem the system is open, the internal dynamics of the baths is ignored, as they are assumed to have fixed thermal spectral densities, and we are interested in the time-independent conductance as a function of the voltage, the Fermi level difference of the external baths.

Despite these differences, there is a strong physical analogy between the two problems [37], since one may interpret the quench problem as follows: The defect can be viewed as an open system coupled to the two halves of the rest of the system. At large times, assuming that a NESS is reached, the particle current coming towards the defect from each side is constant and determined by the particle density produced at quench time far from the defect on the left and right side. Therefore the two halves can be viewed as infinite baths with fixed densities in contact with the defect. It is then reasonable to expect that Landauer's theory is compatible with the semiclassical predictions for the NESS current at the middle in the quench problem. An important difference however is that the effective baths in the quench protocol are not thermal, but described instead by Generalised Gibbs Ensembles corresponding to the homogeneous quenches on the left and right side asymptotically far from the origin. This means in particular that the left/right density imbalance is generally non-zero at all energy levels, in contrast to the case of low-temperature thermal states that is usually considered in Landauer's problem. For the same reason the density imbalance cannot be generally described in terms of voltage, i.e. difference between Fermi levels of thermal distributions.

It is easy to see that our semiclassical expression (13) for the current at the middle is compatible with Landauer's results in the special case of thermal initial densities with different Fermi levels on the two sides. Indeed, changing the integration variable from momentum to energy using $v(k) = dE/dk$, taking into account the double energy level degeneracy and finally setting $\rho_{0L}(E) = f_\beta(E - E_F)$ and $\rho_{0R}(E) = f_\beta(E - E_F - V)$ where $f_\beta(E - E_F)$ is the Fermi-Dirac distribution at temperature $1/\beta$, Fermi level $E_F$ and $V$ is the voltage (Fermi level difference on the two sides), we find that for $V \ll E_F$

$$J \approx 2V \int \frac{dE}{2\pi} T(E) \frac{\partial f_\beta(E - E_F)}{\partial E}, \tag{17}$$

which is the well-known Landauer-Büttiker formula. At arbitrary voltage but zero temperature $1/\beta \to 0$ we obtain

$$J = -2 \int\limits_{E_F}^{E_F + V} \frac{dE}{2\pi} T(E), \quad \text{for } 1/\beta \to 0, \tag{18}$$

which is another well-known result of Landauer's theory.

## 3 Exact analytical derivation of the asymptotics

### 3.1 The quench protocol

We consider a system of free fermions in a one-dimensional lattice of length $L$, assumed for simplicity to be even. We prepare the system in an initial state $\rho_0$ that is diagonal in the local basis with a stepwise inhomogeneity in the density

$$n_0(x) = \nu + \mu \, \text{sign}(x - 1/2), \tag{19}$$

and we let it evolve under the free hopping fermion Hamiltonian with a defect in the hopping located at the origin (precisely, between the sites 0 and 1)

$$H = -\sum_{\substack{x=-L/2+1 \\ x \neq 0}}^{L/2-1} \left( c_x^\dagger c_{x+1} + c_{x+1}^\dagger c_x \right) - j \left( c_0^\dagger c_1 + c_1^\dagger c_0 \right).$$

As can be seen, we have set the bulk hopping $J_b = 1$ and the defect hopping $J_d = j$.

We are interested in the expectation value of the current,

$$J(x,t) = \tfrac{1}{2i} \left( c_x^\dagger c_{x+1} - c_{x+1}^\dagger c_x \right), \tag{20}$$

and the particle density,

$$n(x,t) = c_x^\dagger c_x, \tag{21}$$

in two types of limits: i) at fixed position $x$ and large time $t \to \infty$, and ii) at large times and distances, but keeping the ratio $x/t$ fixed. More generally we are interested in the asymptotics of the two-point correlation function

$$C(x,x';t) = \text{Tr} \left\{ \rho_0 \, e^{+iHt} \, c_x^\dagger c_{x'} \, e^{-iHt} \right\},$$

from which the above and any other observable can be derived. This is because both the initial state and the time evolution are Gaussian in the fermionic modes $c_x, c_x^\dagger$, therefore the problem reduces to a single-particle problem and multi-particle observables can be derived using Wick's theorem.

## 3.2 Post-quench eigenstates

We start by characterising the eigenfunctions $\phi_E(x)$ and corresponding eigenvalues $E$ of the Hamiltonian $H$ in the single particle sector. These are either odd or even with respect to the spatial reflection $x \to 1 - x$, so we can distinguish them with a sign index $\sigma = \pm$

$$\phi_{\sigma,E}(x) = \sigma \phi_{\sigma,E}(1-x). \tag{22}$$

To determine their form, we can focus on their restriction in the right semiaxis ($x \geq 1$) and use the condition that they are eigenvectors of the single particle Hamiltonian together with reflection symmetry in order to find the allowed discrete values of the energy $E$.

In the bulk (anywhere except at the origin) the eigenvector equation becomes

$$\phi_{\sigma,E}(x+1) + \phi_{\sigma,E}(x-1) = -E\phi_{\sigma,E}(x), \quad \text{for all } 1 < x < L/2. \tag{23}$$

This is a recursive equation with general solution

$$\phi_{\sigma,E}(x) = N\cos[k_E(x - \delta_\sigma(k_E))], \text{ (valid for } x > 0), \tag{24}$$

where the new "momentum" parameter $k$ is given as a function of $E$ through

$$E(k) = -2\cos k, \tag{25}$$

which is the dispersion relation. The parameter $\delta_\sigma(k)$ is a phase-shift or "extrapolation length" to be determined by the "initial" conditions of the recursive equation, that is, by the conditions at the origin. From now on we will be using $k$ instead of $E$ to parametrise the eigenfunctions as $\phi_\sigma(k;x)$.

At the origin the eigenvector equation becomes

$$\phi_\sigma(k;2) + j\phi_\sigma(k;0) = -E(k)\phi_\sigma(k;1),$$

which, together with the reflection symmetry (22) that can be written as a condition on the middle site values $\phi_\sigma(k;0) = \sigma \phi_\sigma(k;1)$, gives

$$\phi_\sigma(k;2) = -[E(k)+\sigma j]\phi_\sigma(k;1).$$

This plays the role of "initial" condition for the above recursive equation (23). By requiring that the general solution (24) satisfies it, we find that $\delta_\sigma(k)$ must satisfy the equation

$$e^{2ik\delta_\sigma(k)} = e^{ik}\frac{1-\sigma j e^{ik}}{\sigma j - e^{ik}}, \tag{26}$$

which defines $\delta_\sigma(k)$.

At the right boundary $x = L/2$, the eigenfunctions satisfy the equation

$$\phi_\sigma(k;L/2-1) = -E(k)\phi_\sigma(k;L/2),$$

which can be written as

$$\cos k\,(L/2+1-\delta_\sigma(k)) = 0, \tag{27}$$

and constitutes the quantisation condition determining the discrete values of the parameter $k$ (or $E$, through (25)).

Lastly, the normalisation factor $N$ turns out to be

$$N_\sigma(k) = \left(L/2 + \frac{1-j\sigma\cos k}{1+j^2-2j\sigma\cos k}\right)^{-1/2} \to \sqrt{2/L} \quad \text{for } L \to \infty. \tag{28}$$

For $j > 1$ there are also two bound states, one even and one odd, that are localised at the defect and correspond to imaginary $k$. In the thermodynamic limit $L \to \infty$, they are given by

$$\phi_{b,+}(x) = N_b\, e^{-\kappa|x-1/2|}, \tag{29}$$

$$\phi_{b,-}(x) = N_b\, (-1)^x e^{-\kappa|x-1/2|}, \tag{30}$$

with $\kappa = \log j$ and the corresponding energies

$$E_{b,\pm} = \mp\left(j+j^{-1}\right). \tag{31}$$

### 3.3 Calculation of the two point correlation function

We will now calculate the two point correlation function in the thermodynamic limit and derive its asymptotics at large times, using a mathematically rigorous approach [57]. Let us focus on the case $j < 1$, in which there are no bound states. The calculation in the case $j > 1$ is analogous and is outlined in App. A.

The time evolution of the two-point correlation function can be expressed in terms of the eigenfunctions as

$$
\begin{aligned}
C(x,x';t) &= \text{Tr}\left\{\rho_0\, e^{+iHt}\, c_x^\dagger c_{x'}\, e^{-iHt}\right\}\\
&= \sum_{\sigma,\sigma'}\sum_{k,k'}\phi_\sigma(k;x)\phi_{\sigma'}(k';x')e^{i(E(k)-E(k'))t}\,\text{Tr}\left\{\rho_0\,\hat c_{\sigma,k}^\dagger \hat c_{\sigma',k'}\right\}\\
&= \sum_{y,y'}\sum_{\sigma,\sigma'}\sum_{k,k'}\phi_\sigma(k;x)\phi_{\sigma'}(k';x')\phi_\sigma(k;y)\phi_{\sigma'}(k';y')e^{i(E(k)-E(k'))t}\,\text{Tr}\left\{\rho_0\,c_y^\dagger c_{y'}\right\}\\
&= \sum_{y,y'}G^*(x,y;t)G(x',y';t)C_0(y,y'), \tag{32}
\end{aligned}
$$

where

$$G(x,y;t) = \sum_\sigma \sum_k \phi_\sigma(k;x)\phi_\sigma(k;y)e^{-iE(k)t} \tag{33}$$

is the free fermion propagator, and

$$C_0(y,y') = \mathrm{Tr}\left\{\rho_0\, c_y^\dagger c_{y'}\right\}$$

is the initial correlation function. For our initial state (19) with only diagonal correlations in the local basis, this is given by

$$C_0(y,y') = \delta_{y,y'}n_0(y),$$

and using the stepwise form of the initial density (19), we find

$$C(x,x';t) = \nu \sum_{y=-L/2+1}^{L/2} G^*(x,y;t)G(x',y;t)$$

$$+ \mu \sum_{y=1}^{L/2}\left(G^*(x,y;t)G(x',y;t) - G^*(x,1-y;t)G(x',1-y;t)\right). \tag{34}$$

The first term is irrelevant for transport, since by straightforward calculation using the completeness of the eigenstates we find

$$\sum_{y=-L/2+1}^{L/2} G^*(x,y;t)G(x',y;t) = \delta_{x,x'}.$$

Using the reflection symmetry of the eigenstates, the second term can be written as

$$\sum_{y=1}^{L/2}\left(G^*(x,y;t)G(x',y;t) - G^*(x,1-y;t)G(x',1-y;t)\right)$$

$$= 2\sum_\sigma \sum_{k,k'}\phi_\sigma(k;x)\phi_{-\sigma}(k';x')\sum_{y=1}^{L/2}\phi_\sigma(k;y)\phi_{-\sigma}(k';y)e^{i\left(E(k)-E(k')\right)t}. \tag{35}$$

We calculate the spatial sum of the last expression and eliminate $L$ using the quantisation condition (27) in the form $e^{ik(L/2+1-\delta_\sigma(k))} = -e^{-ik(L/2+1-\delta_\sigma(k))}$ in order to perform the thermodynamic limit $L \to \infty$ easier

$$\sum_{y=1}^{L/2}\phi_\sigma(k;y)\phi_{-\sigma}(k';y) = \frac{N_\sigma(k)N_{-\sigma}(k')}{2} \times$$

$$\times \frac{1}{(\cos k - \cos k')}\left[\cos k\,\delta_\sigma(k)\cos k'\left(1-\delta_{-\sigma}(k')\right) - \cos k\left(1-\delta_\sigma(k)\right)\cos k'\,\delta_{-\sigma}(k')\right]. \tag{36}$$

Having eliminated $L$, the summand of (35) is now a function of variables $k,k'$ that, in the thermodynamic limit, spread uniformly in the interval from 0 to $\pi$, therefore we can write the sums over $k,k'$ as integrals. However, since the functions $\delta_\sigma(k), \delta_{-\sigma}(k')$ are implicitly known and defined in terms of the discrete values of $k,k'$, we do not know if they converge uniformly to smooth functions in this limit. Luckily, this problem can be circumvented easily:

if we multiply with $\phi_\sigma(k;x)\phi_{-\sigma}(k';x')$ and use the defining equation (26) of these implicitly known functions, we can eliminate them completely

$$\phi_\sigma(k;x)\phi_{-\sigma}(k';x')\sum_{y=1}^{L/2}\phi_\sigma(k;y)\phi_{-\sigma}(k';y)$$

$$= \sigma j N_\sigma^2(k)N_{-\sigma}^2(k')\frac{\sin k \sin k' \left(\sin kx + \sigma j \sin k(1-x)\right)\left(\sin k'x' - \sigma j \sin k'(1-x')\right)}{(\cos k - \cos k')(1+j^2-2\sigma j \cos k)(1+j^2+2\sigma j \cos k')}.$$

Notice that, while the original summand of (35) is well-defined for all discrete values of $k, k'$ and any finite $L$, the continuous function to which it converges in the thermodynamic limit exhibits a pole at $k = k'$. This pole, which reflects precisely the inhomogeneity of the initial state, is going to play a crucial role in the derivation of the NESS, as it turns out to completely determine the combined thermodynamic and large distance and time asymptotics. Its presence prevents us from writing the thermodynamic limit of the sums directly as integrals, since we need to find the correct contour prescription for how to pass around the pole. This can be done using a standard complex analysis trick, described in more detail in Appendix B.1: we introduce a meromorphic function with poles of unit residue at the discrete values of $k'$ that we sum over, so that we can write the sum over $k'$ as a contour integral along straight lines just above and below the real axis, but paying attention to subtract the residue of the additional pole at $k' = k$. Then we can safely take the thermodynamic limit, in which only the contribution of the extra pole and of one of the above straight line integrals survives.

Using this trick (equation (64) of the appendix) and substituting into (34), we finally obtain an exact formula for the thermodynamic limit of the correlation function

$$\lim_{L\to\infty} C(x,x';t) = \nu\delta_{x,x'}$$

$$+ 2\mu\sum_\sigma \sigma j \int_0^\pi \frac{dk}{\pi}\left[\int_{0-i\epsilon}^{\pi-i\epsilon}\frac{dk'}{\pi}-i\left(1-\frac{i\sigma(1-j^2)}{2j\sin k}\right)\operatorname*{Res}_{k'=k}\right]$$

$$\times \frac{\sin k \sin k' \left(\sin kx + \sigma j \sin k(1-x)\right)\left(\sin k'x' - \sigma j \sin k'(1-x')\right)}{(\cos k - \cos k')(1+j^2-2\sigma j \cos k)(1+j^2+2\sigma j \cos k')}e^{i\left(E(k)-E(k')\right)t}, \quad (37)$$

where the integral over $k'$ runs along a contour from 0 to $\pi$ slightly below the real axis.

The last expression holds only for $x, x' > 0$, since in the above calculation we used the right half side expression of the eigenfunctions $\phi_\sigma(k;x)$. The explicit expression in different spatial regions can be deduced from the above, as already mentioned, by exploiting the reflection symmetry with respect to the origin.

## 3.4 Large time fixed position density and current

In the large time limit $t \to \infty$ at fixed positions $x, x'$ the contour integral over $k'$ in (37) vanishes, since from the dispersion relation (25) we find that the exponent of $e^{-iE(k')t}$ has a negative real part $-(dE/dk')\epsilon t = -\sin k' \epsilon t < 0$ for all $k'$ from 0 to $\pi$. Therefore only the residue of the pole at $k' = k$ survives and we find

$$\lim_{t\to\infty}\lim_{L\to\infty} C(x,x';t) = \nu\delta_{x,x'} + \mu\int_0^\pi \frac{dk}{\pi}\frac{1}{(1+j^4-2j^2\cos 2k)}\times$$

$$\left\{4ij^2\sin^2 k \sin k(x-x') - (1-j^2)\left[(1-j^2)\cos k(x-x') - \cos k(x+x') + j^2\cos k(x+x'-2)\right]\right\}$$

$$\text{(valid for } x, x' > 0). \quad (38)$$

Notice that the multiple sums over eigenstates and the sum over all lattice sites in the original expression (34) have been reduced to a single integral over momenta in the combined thermodynamic and large time limit.

The last expression (38) for the correlation function determines completely the NESS that forms at large times. In particular we find the large time value of the current at the origin as a function of the defect strength

$$J_{\text{NESS}} = \lim_{t \to \infty} J(t) = -\frac{2\mu}{\pi} \int_0^\pi \mathrm{d}k \frac{2j^2 \sin^3 k}{(1 + j^4 - 2j^2 \cos 2k)}$$
$$= -\frac{4\mu}{\pi} \left[ 1 + \frac{(1 - j^2)^2}{2j(1 + j^2)} \log\left(\frac{|1 - j|}{1 + j}\right) \right] \qquad \text{(valid for } j < 1\text{).} \qquad (39)$$

For $j > 1$, as shown in App. A, the large time current is given by the above expression plus additive corrections

$$\delta J(x, t) = -2\mu \frac{(1 + j)^2 (1 - j)^2}{j(1 + j^2)} (-1)^x j^{-|2x-1|-1} \sin 2(j + 1/j)t, \qquad (40)$$

which are due to the presence of bound state excitations. We see that the bound states induce persistent oscillations in time at frequency $2(j + 1/j)$ equal to their energy and which decay exponentially with the distance from the origin at a characteristic length $1/(2 \log j)$.

Notice the symmetry of the expression (39) under the discrete transformation $j \to 1/j$. This reveals an interesting strong-weak duality which, as we will see in the following, is manifest in all results describing the asymptotic behaviour of the system and which is neither present nor obvious in the microscopic description of the problem. As anticipated in Sec. 2, the reason for the emergence of this symmetry is that the asymptotic behaviour can be fully determined from the quasiparticle reflection or transmission probabilities that characterise the defect and which for the hopping defect turn out to be symmetric under inversion of the defect strength.

Another observation we can derive from the expression (38) is that for $j < 1$ the current $J(x, t)$ at fixed position and large time is independent of the position

$$\lim_{t \to \infty} J(x, t) = \lim_{t \to \infty} J(0, t) \quad \text{(valid for } j < 1\text{).}$$

This is because the expression for $\lim_{t \to \infty} J(x, t)$ is given by the same integral as $\lim_{t \to \infty} J(0, t)$ independently of the position $x$.

The large time density $n(x, t)$, on the other hand, exhibits a stepwise inhomogeneity in space but is otherwise independent of the position. More specifically, we find that for $x \geq 1$

$$\lim_{t \to \infty} n(x, t) = \nu + \mu \int_0^\pi \frac{\mathrm{d}k}{\pi} \frac{(1 - j^2) \left[ 1 - j^2 - \cos 2kx + j^2 \cos 2k(x - 1) \right]}{(1 + j^4 - 2j^2 \cos 2k)}, \qquad (41)$$

while for $x \leq 0$, using the reflection symmetry we have

$$\lim_{t \to \infty} n(x, t) = 2\nu - \lim_{t \to \infty} n(1 - x, t).$$

The calculation of the above integral is a standard application of residue theorem: we first express it in terms of the complex variable $z = \mathrm{e}^{2\mathrm{i}k}$, separate the integrand in powers of $z^x$ and note that the various terms have poles at either $z = j^2$ or $z = 1/j^2$ or both, which are located inside or outside of the unit circle depending on whether $j < 1$ or $j > 1$. In the present case $j < 1$, it turns out that these poles do not contribute to the above expression.

After some algebra, we finally find that for $j < 1$ the NESS density exhibits a simple step but is otherwise independent of the position

$$\lim_{t \to \infty} n(x, t) = \nu + \text{sign}(x - 1/2) \mu \frac{|1 - j^2|}{1 + j^2} \qquad \text{(valid for } j < 1\text{).} \qquad (42)$$

For $j > 1$, as shown again in App. A, the density is given by the above expression plus additive corrections

$$\delta n(x,t) = \mu \, j^{-|2x-1|-1} \left[ \text{sign}(x-1/2)(1-j^2) + \frac{(1+j)^2(1-j)^2}{(1+j^2)}(-1)^x \cos 2(j+1/j)t \right],$$
(43)

i.e. apart from a steplike inhomogeneity at large distances, there are also corrections that decay exponentially with the distance from the origin together with persistent oscillations in time that also decay away from the origin. In this case the large time limit does not exist strictly speaking, due to the presence of the oscillatory terms.

The density difference on the left and on the right of the defect and far from it, is given by

$$\Delta n_{\text{NESS}} = \lim_{R\to\infty} \lim_{t\to\infty} (n(+R,t) - n(-R+1,t)) = 2\mu \frac{|1-j^2|}{1+j^2}.$$
(44)

We observe that the asymptotic NESS density difference is proportional to and has the same sign as the initial density difference $\Delta n_0 = n_0(x > 0) - n_0(x < 0) = 2\mu$, that is the defect preserves part of the initial density difference. Note that, even though for $j < 1$ the difference on the two sides far from the origin is equal to the difference between the sites 0 and 1, this is not true for $j > 1$ where, as already mentioned, there are corrections close to the origin that result in a time-averaged density difference equal to

$$\Delta n_d = \bar{n}(1) - \bar{n}(0) = -2\mu \frac{(j^2-1)}{j^2(1+j^2)} \quad \text{(for } j > 1\text{)},$$
(45)

where the bars denote infinite time averages. Note that this difference has the opposite sign of the initial density step.

## 3.5 Large distance and time asymptotics of density and current

Apart from the asymptotics at fixed position $x$ and large times $t$, we can also derive the asymptotics at large times and distances from the origin keeping their ratio $\xi = x/t$ fixed. To this end we go back to the expression (37) for the correlation function in the thermodynamic limit, expand the integrand in terms containing exponentials of the form $e^{\pm ik'x - iE(k')t}$ and derive the asymptotics of the $k'$-integral of each term separately. This can be done by deforming if necessary the integration contour slightly above the real $k'$-axis to make sure that it passes only through regions where the exponent has a negative real part and therefore the new integral decays in the considered limit. The necessary contour deformation depends on the value of $x/t$. If the pole of the integrand at $k' = k$ is crossed during the contour deformation, then we have to take into account the contribution of its residue. This complex analysis method is described in more detail in Appendix B.2.

Using this method (equation (65) of the appendix), we finally find that the density $n$ as a function of the rescaled variable $x/t$ is

$$n_\infty(x/t) = \lim_{\substack{x,t\to\infty \\ x/t:\ \text{fixed}}} n(x,t) = \nu - \mu F(j; x/t),$$

where $F(j; \xi)$ is the following piecewise defined function

$$F(j; \xi) =$$

$$= \begin{cases} +1 & \text{for } \xi < -2 \\ \left( \frac{|1-j^2|}{1+j^2} \right) - \frac{2}{\pi} \arctan \left( \frac{\xi}{\sqrt{4-\xi^2}} \right) + \frac{2}{\pi} \left( \frac{1-j^2}{1+j^2} \right) \arctan \left( \frac{1+j^2}{1-j^2} \frac{\xi}{\sqrt{4-\xi^2}} \right) & \text{for } -2 < \xi < 0 \\ -\left( \frac{|1-j^2|}{1+j^2} \right) - \frac{2}{\pi} \arctan \left( \frac{\xi}{\sqrt{4-\xi^2}} \right) + \frac{2}{\pi} \left( \frac{1-j^2}{1+j^2} \right) \arctan \left( \frac{1+j^2}{1-j^2} \frac{\xi}{\sqrt{4-\xi^2}} \right) & \text{for } 0 < \xi < +2 \\ -1 & \text{for } \xi > +2. \end{cases} \tag{46}$$

Note that the above function satisfies the expected properties: It equals $\pm 1$ outside of the light-cone $|\xi| = |x|/t > v_{\text{max}} = 2$ on the left/right side respectively, reducing to the initial left/right half side density values. For $j = 1$ i.e. in the absence of defect, it reduces to

$$F(1; \xi) = \begin{cases} +1 & \text{for } \xi < -2 \\ -\frac{2}{\pi} \arcsin(\xi/2) & \text{for } -2 < \xi < 2 \\ -1 & \text{for } \xi > +2, \end{cases} \tag{47}$$

which describes the rescaled density profile for an inhomogeneous initial state evolving under the homogeneous defectless Hamiltonian, derived in [10]. For $j = 0$ or $\infty$, i.e. when the defect is impenetrable and keeps the two halves of the system completely disconnected from each other, $F$ becomes

$$F(0; \xi) = F(\infty; \xi) = \begin{cases} +1 & \text{for } \xi < 0 \\ -1 & \text{for } \xi > 0, \end{cases}$$

that is, there is no transport at all and the density remains everywhere equal to the initial value.

For any $j \neq 1$ on the contrary, it exhibits a discontinuity at $\xi = 0$ which is the most characteristic effect of the presence of the defect. Note that the value of the discontinuity jump

$$\lim_{\xi \to 0^+} n_\infty(\xi) - \lim_{\xi \to 0^-} n_\infty(\xi) = \Delta n = 2\mu \frac{|1 - j^2|}{1 + j^2}$$

is equal to the value $\Delta n_{\text{NESS}}$ in (44), showing that these two different limits match.

The asymptotic profile of the current $J$ as a function of the rescaled variable $x/t$ can be calculated directly from the density profile $n(x/t)$ using the continuity equation $\partial_t n = -\partial_x J$ which gives $J_\infty(x/t) = \int_{-\infty}^{x/t} \xi \, n'_\infty(\xi) d\xi$

$$J_\infty(x/t) = \lim_{\substack{x,t \to \infty \\ x/t: \text{fixed}}} J(x, t) = -\mu G(j; x/t),$$

where the function $G(j; \xi)$ is defined as

$$G(j; \xi) = \int_{-\infty}^{\xi} \xi' \, \partial_{\xi'} F(j; \xi') d\xi'$$

$$= \begin{cases} \frac{2}{\pi} \left[ \sqrt{4 - \xi^2} - \frac{(1-j^2)^2}{j(1+j^2)} \text{arctanh} \left( \frac{j}{1+j^2} \sqrt{4 - \xi^2} \right) \right] & \text{for } -2 < \xi < 2 \\ 0 & \text{otherwise.} \end{cases} \tag{48}$$

For $j = 1$ we recover the known result for the case of absence of the defect

$$G(1; \xi) = \begin{cases} \frac{2}{\pi} \sqrt{4 - \xi^2} & \text{for } -2 < \xi < 2 \\ 0 & \text{otherwise,} \end{cases}$$

and for $j = 0$ or $\infty$, we have $G(0, \xi) = G(\infty, \xi) = 0$, as expected. Note also that the current is continuous as $|x|/t \to 0$ and its value is given by

$$G(j; 0) = \frac{4}{\pi} \left[ 1 + \frac{(1 - j^2)^2}{2j(1 + j^2)} \log \left( \frac{|1 - j|}{1 + j} \right) \right],$$

that gives the same value (39) found earlier for the large time current at the origin $J_{\text{NESS}}$, which means that the two limits give the same result, as for the density profile.

Lastly note that the functions $F(j; \xi)$ and $G(j; \xi)$ are symmetric under inversion of the defect strength $j \to 1/j$, i.e. $F(j; \xi) = F(1/j; \xi)$ and $G(j; \xi) = G(1/j; \xi)$. This shows that the strong/weak defect duality pointed out earlier for the NESS expectation values (4) and (44) is manifest also in the large distance and time asymptotics of the density and consequently the current.

## 4 Comparison with numerics

We will now compare our analytical results with numerical data. Since the defect considered here is non-interacting, the full quench dynamics can be obtained efficiently by exact diagonalisation[1]. The large time and distance asymptotics can be observed by considering a sufficiently large finite system and long times up to some order smaller than the system size divided by the maximum group velocity, in order to exclude finite size effects.

In Fig. 4 the analytical results (39) and (44) for the NESS current and density difference are plotted as functions of the defect strength $j$ together with numerics. The numerical data are extracted from the current and density profiles at the largest time reached in the computation ($t = 50$). For $j > 1$, in order to subtract the time oscillation corrections (40) and (43) close to the defect, we average over a sufficiently large time window (from $t = 40$ to $50$) to integrate them out. For the estimation of the NESS density difference we also apply a linear fit on the spatial profiles in order to extrapolate the values on the left and right from the origin, eliminating as much as possible the exponentially decaying corrections.

There are some small deviations between analytics and numerics for the density difference only for $j > 1$ and close to 1. These are exactly due to the above finite distance effects and, more precisely, due the fact that they spread over a characteristic length that diverges as $j \to 1^+$, meaning that at any finite time there is some mixing between these features and the asymptotic form of the profile. For this reason, data extrapolation from linear fitting is less effective in this region and the estimates of $\Delta n_{\text{NESS}}$ are distorted towards $\Delta n_d$ given in (45) which has opposite sign. Except for these clearly understood deviations, we observe perfect agreement between analytical and numerical results. Note also the reflection symmetry of the curve with respect to the vertical line at $j = 1$ in logarithmic-linear plot, which demonstrates the emergent strong-weak duality symmetry $j \to 1/j$.

The analytical expressions (46) and (48) are plotted against numerical data in Fig. 5 and 6 for several values of $j \leq 1$ and $j \geq 1$ respectively. The numerical data correspond to the maximum time reached in our computation (again $t = 50$).

Notice the oscillatory behaviour of the numerics close to the fronts. This is shown in more detail in Fig. 7 where numerical data for the current profile close to the right front are plotted

---

[1] In our plots we have actually used data produced by tDMRG simulations as a special case of an extensive numerical study of more general defects including interactions for which tDMRG is suitable. For the present non-interacting defect problem however, the use of tDMRG is not the optimal choice as the exact diagonalisation is more efficient. Nevertheless, we have verified the accuracy of our tDMRG data by comparison with exact diagonalisation for certain parameter values. Moreover, the excellent match between our analytical and numerical computations reported below is as well a good unbiased test for the accuracy of tDMRG method.

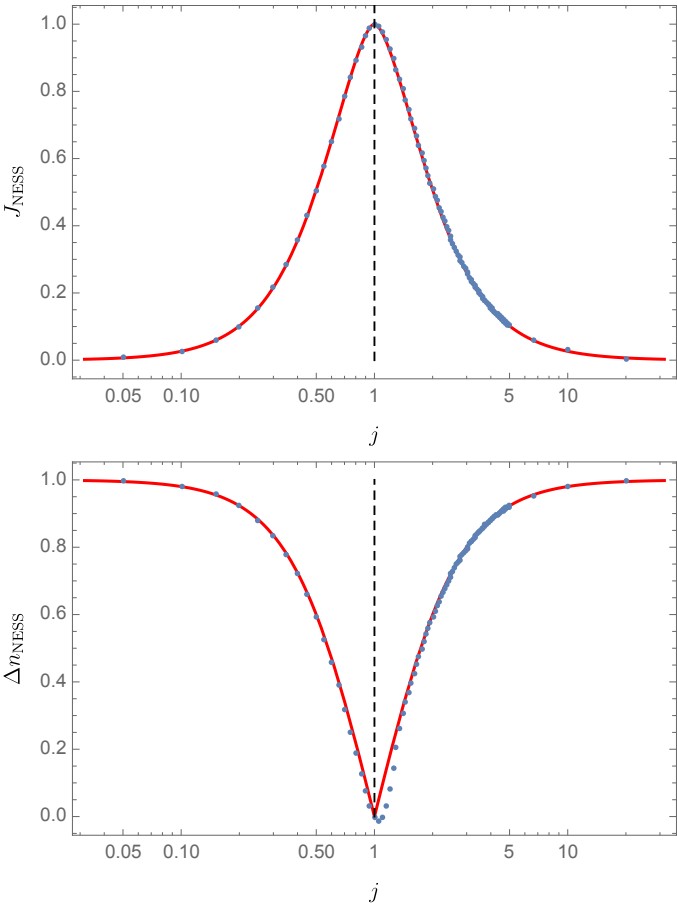

Figure 4: Log-linear plots of the NESS current at the origin $J_{\text{NESS}}$ (*top*) and density difference far on the left and on the right from the defect $\Delta n_{\text{NESS}}$ (*bottom*) as functions of the defect strength $j$ (ratio of defect over bulk hopping). Current and density units are $-4\mu/\pi$ and $2\mu$ respectively, with $2\mu$ the size of the initial density step. Analytical results (red curves) given by (39) and (44) vs. estimates extracted from numerics (blue dots).

for several different times together with the analytical curve. These features of the profile are described by the Airy function [16, 18] and are finite time corrections, since even though they broaden in space as time increases, their width scales slower than linearly with time and therefore shrinks to a point as $t \to \infty$ when plotting in terms of the rescaled variable $x/t$. This is the reason why this effect is not present in the asymptotic expressions (46) and (48). Also notice the oscillatory and exponentially damped deviations of the numerics close to the defect for $j > 1$ (Fig. 6), which, as already explained, are restricted to finite distances from the origin and therefore also disappear in the rescaled variable plots when $t \to \infty$.

With the exception of these expected corrections due to finite distance or time effects, the analytical curves match perfectly with the numerics. In Fig. 8 later on we perform the same comparison between analytics and numerics for a different type of non-interacting defect, observing once again perfect agreement.
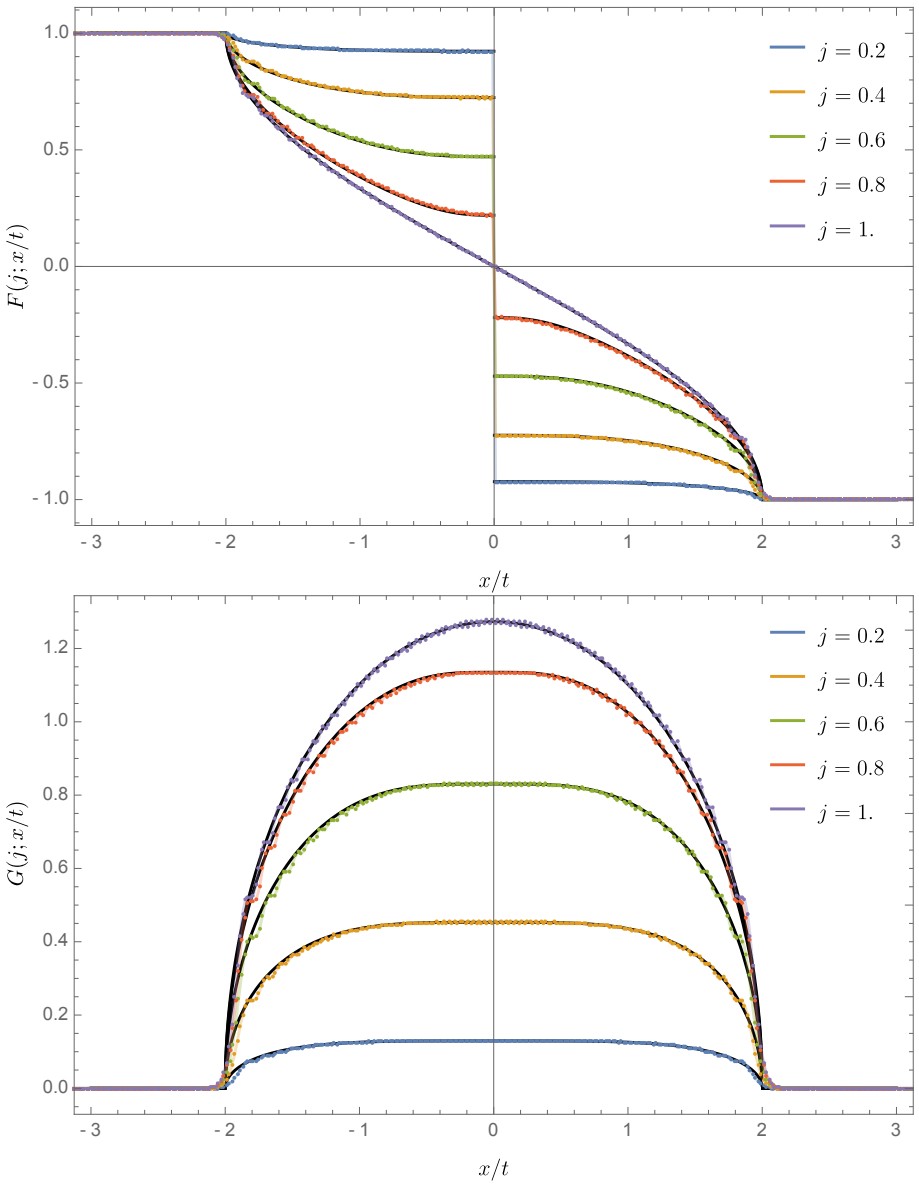

Figure 5: Asymptotics of the density $n_\infty(x/t)$ (*top*) and current $J_\infty(x/t)$ (*bottom*) as given by the scaling functions $F(j; x/t)$ and $G(j; x/t)$ respectively for different values of $j \leq 1$. Coloured lines and points: numerical data at the maximum time reached ($t = 50$), black lines: analytical expressions from (46) and (48).

## 5 General defect: asymptotics from the scattering matrix

It is easy to verify that the above analytical results (46) and (48) for the asymptotic density and current profiles are identical to the semiclassical predictions (9), (10) and (12) once the reflection and transmission probabilities are substituted by their explicit form (15) and (16) for a hopping defect. We will now show that this is generally true for any type of non-interacting and particle number preserving defects. To this end we are going to generalise our analytical calculation using the scattering matrix that completely characterises such a defect. Scattering matrix based analytical approaches for the study of quantum transport in presence of a defect have been used in the context of quantum wires or quantum dots coupled to infinite baths [58–61] and recently for the study of light cone effects after a local quantum quench [62].

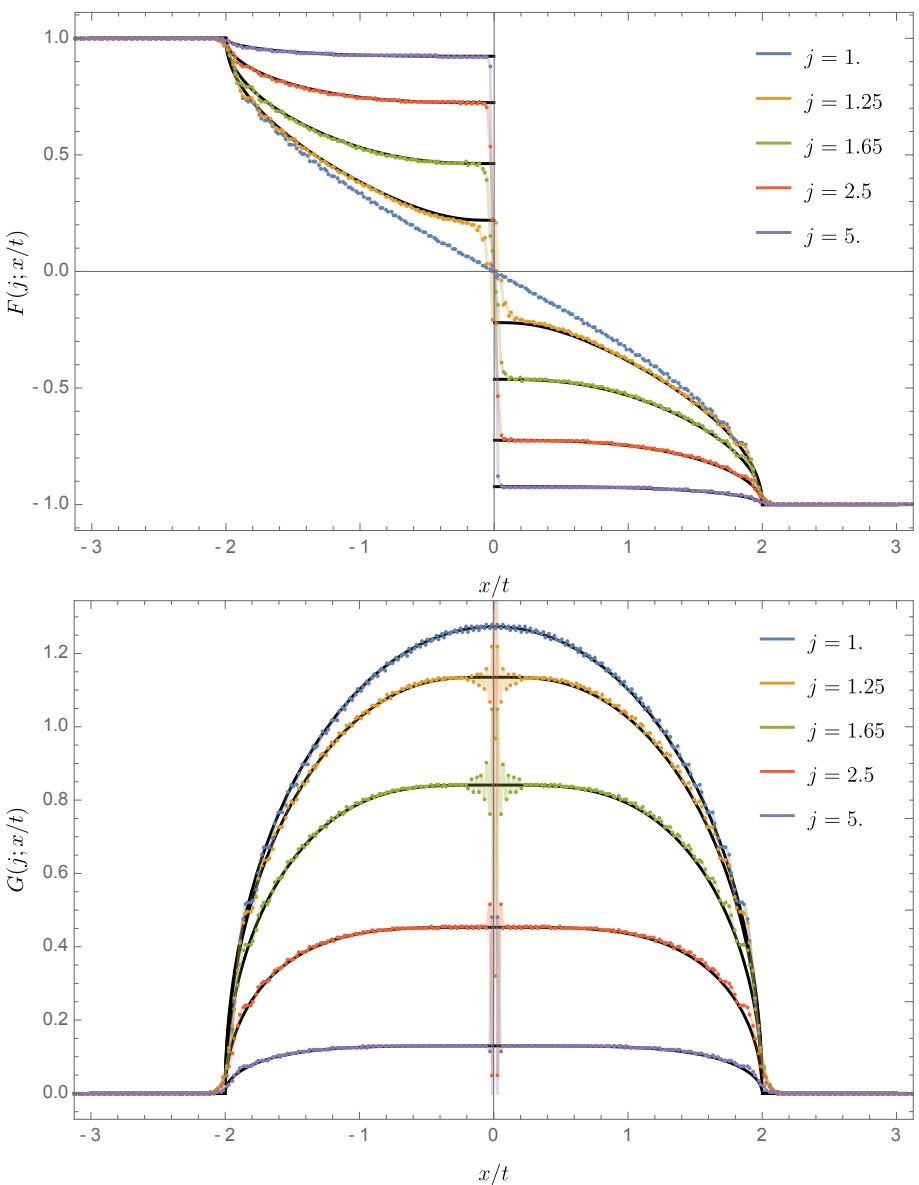

Figure 6: Asymptotics of the density $n_\infty(x/t)$ (*top*) and current $J_\infty(x/t)$ (*bottom*) as given by the scaling functions $F(j;x/t)$ and $G(j;x/t)$ respectively for different values of $j \geq 1$. Coloured lines and points: numerical data at the maximum time reached ($t = 50$), black lines: analytical expressions from (46) and (48).

We consider a general non-interacting defect $V_d$, located in a finite spatial interval $[-\ell/2 + 1, +\ell/2]$ centred at the origin (here $x = 1/2$), focusing for simplicity on reflection symmetric defects

$$H = -\sum_{\substack{x=-L+1 \\ x \neq 0}}^{L-1} c_x^\dagger c_{x+1} + V_d.$$

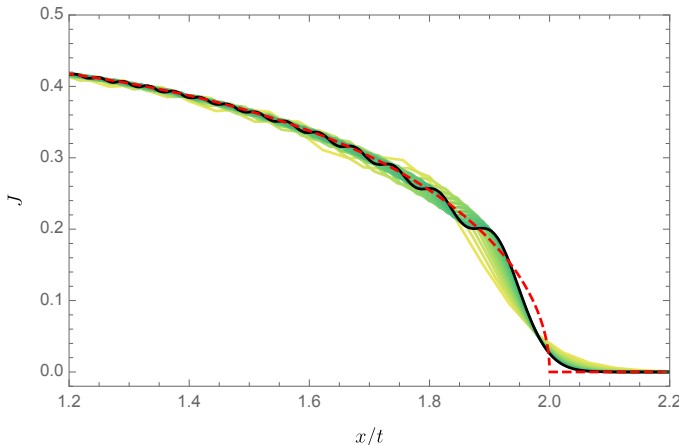

Figure 7: Current profile $J$ as a function of $x/t$ close to the right front $x/t = 2$ for $j = 0.4$. Full lines: numerical data for different times $t = 25$ to $95$ (yellow to green) at steps equal to $5$ and $t = 100$ (black line), dashed red line: analytical asymptotic expression at $t \to \infty$ from (48).

The eigenstates are plane waves with a phase shift between the left and right side of the defect, which can be written as

$$\phi(k; x) \propto \left(A_k e^{ik(x-1/2)} + B_k e^{-ik(x-1/2)}\right)\Theta(x < -\ell/2)$$
$$+ [\text{unknown form inside defect interval}] + \left(C_k e^{ik(x-1/2)} + D_k e^{-ik(x-1/2)}\right)\Theta(x > 1 + \ell/2). \tag{49}$$

The scattering or S-matrix is defined through the relation

$$\begin{pmatrix} C \\ B \end{pmatrix} = S \begin{pmatrix} A \\ D \end{pmatrix},$$

where

$$S = \begin{pmatrix} S_{11} & S_{12} \\ S_{21} & S_{22} \end{pmatrix} = \begin{pmatrix} t_L & r_R \\ r_L & t_R \end{pmatrix},$$

and $r_{L/R}, t_{L/R}$ are the reflection and transmission probability amplitudes for a particle incident from the left ($D = 0$) or right ($A = 0$) respectively. For defects symmetric under spatial reflections, the S-matrix is a symmetric matrix with only two complex parameters

$$S = \begin{pmatrix} t & r \\ r & t \end{pmatrix}.$$

From the requirement of particle conservation, the S-matrix must always be a unitary matrix, i.e. $S^\dagger S = I$, which means that $r, t$ satisfy the relations

$$|r|^2 + |t|^2 = 1, \tag{50}$$
$$r t^* + r^* t = 0. \tag{51}$$

As always the reflection and transmission probabilities are defined as the norm squares of the corresponding amplitudes

$$R = |r|^2,$$
$$T = |t|^2.$$

Note also the useful formula

$$r^2 = -t^2 \frac{R}{T}, \tag{52}$$

that can be found from the above. The eigenvalues of the S-matrix are the unitary numbers $t \pm r$.

In the symmetric defect case, the eigenfunctions are either odd or even

$$\phi_\pm(k; x \le -\ell/2) = \pm \phi_\pm(k; 1 - x > +\ell/2).$$

We therefore find additional relations between the coefficients $A, B, C, D$

$$\sigma = + : \quad B = C, A = D,$$
$$\sigma = - : \quad B = -C, A = -D.$$

We can choose to parametrise the solutions (49) with respect to the right side coefficients $C, D$. Moreover it is convenient to rewrite them in trigonometric form as

$$\phi_\sigma(k; x) = N \cos k(x - \delta_\sigma(k)), \text{ (valid for } x \ge 0),$$

as in the previously studied case of a hopping defect. The energy eigenvalues are as always given by

$$E(k) = -2 \cos k,$$

and the normalisation factor

$$N = \sqrt{2/L}, \quad \text{for } L \to \infty,$$

while $\delta_\sigma(k)$ is an "extrapolation length" or "phase shift" parameter to be determined by the matching conditions $\begin{pmatrix} C \\ B \end{pmatrix} = S \begin{pmatrix} A \\ D \end{pmatrix}$ at the defect. It can be easily expressed in terms of $D, C$ or $r, t$ as

$$\begin{aligned} e^{2ik\delta_\sigma(k)} &= e^{ik} \frac{D_{\sigma,k}}{C_{\sigma,k}} \\ &= e^{ik} \frac{1}{r(k) + \sigma t(k)} \\ &= e^{ik} (r^*(k) + \sigma t^*(k)). \end{aligned} \tag{53}$$

The quantisation conditions can be still written in terms of $\delta_\sigma(k)$ as

$$\cos k (L/2 + 1 - \delta_\sigma(k)) = 0. \tag{54}$$

Having expressed the eigenstates in the same form as in the previous special case, the calculation of the correlation function and the asymptotic profiles can be done by repeating the method of Sec. 3.3 and 3.4. We thus find

$$\lim_{t \to \infty} \lim_{L \to \infty} C(x, x'; t) = \nu \delta_{x,x'}$$

$$\begin{aligned} + \mu \int_0^\pi \frac{dk}{\pi} \frac{1}{t^2(k) - r^2(k)} \Big[ r^2(k) \cos k(x - x') + it^2(k) \sin k(x - x') \\ + \tfrac{1}{2} r(k)\big(r^2(k) - t^2(k) + 1\big) \cos k(x + x' - 1) \\ + \tfrac{1}{2} ir(k)\big(r^2(k) - t^2(k) - 1\big) \sin k(x + x' - 1) \Big] \quad \text{(valid for } x, x' > 1 + \ell/2), \end{aligned}$$

which can be equivalently written in terms of the phase shifts $e^{2ik\delta_\pm(k)}$ as

$$
\lim_{t\to\infty}\lim_{L\to\infty} C(x,x';t) = \nu\delta_{x,x'}
$$
$$
-\mu\int_0^\pi \frac{dk}{4\pi}\sum_{\sigma=+,-}\left[e^{-ikx+ikx'} + e^{ikx+ikx'-2ik\delta_\sigma(k)} + e^{-ikx-ikx'+2ik\delta_\sigma(k)}\right.
$$
$$
\left. + e^{ikx-ikx'-2ik\delta_\sigma(k)+2ik\delta_{-\sigma}(k)}\right] \qquad \text{(valid for } x,x' > 1+\ell/2\text{).}
$$

From these expressions we can derive formulas for the NESS current and density difference far on the left and right of the defect, taking into account that potential presence of bound states or singularities of the S-matrix would only result in corrections that decay exponentially with the distance from the defect and therefore vanish far from it. Using (52), it turns out that both of these values can be expressed in a very simple form in terms of only the reflection $R(k)$ or transmission probability $T(k) = 1 - R(k)$

$$
\Delta n_{\text{NESS}} = \lim_{x\to\infty}\lim_{t\to\infty}(n(x,t) - n(1-x,t)) = 2\mu\int_0^\pi \frac{dk}{\pi}R(k), \tag{55}
$$

$$
J_{\text{NESS}} = \lim_{|x|\to\infty}\lim_{t\to\infty} J(x,t) = -2\mu\int_0^\pi \frac{dk}{\pi}T(k)\sin k. \tag{56}
$$

Lastly we can derive the density and current asymptotics at large times and distances, which are also expressible in terms of $T(k)$ only

$$
n_\infty(x/t) = \nu + \mu\,\text{sign}(x/t)\left[1 - \int_0^\pi \frac{dk}{\pi}T(k)\Theta(2\sin k - |x/t|)\right], \tag{57}
$$

$$
J_\infty(x/t) = -2\mu\int_0^\pi \frac{dk}{\pi}T(k)\Theta(2\sin k - |x/t|)\sin k. \tag{58}
$$

As already mentioned, we can verify that all above expressions reduce to the previously derived formulas for the special case of hopping defect. The reflection and transmission probability amplitudes are

$$
r(k) = -\frac{(1-j^2)e^{ik}}{1-j^2e^{2ik}}, \tag{59}
$$

$$
t(k) = \frac{(1-e^{2ik})j}{1-j^2e^{2ik}}, \tag{60}
$$

resulting in the corresponding probabilities (15) and (16). Substituting $r(k), t(k)$ from (59) and (60) we recover all previously found formulas for the asymptotics of the correlation function, density and current.

As an additional verification test we now consider a different type of non-interacting defect, a density defect at the sites 0 and 1

$$
H = -\sum_{x=-L/2+1}^{L/2-1}\left(c_x^\dagger c_{x+1} + c_{x+1}^\dagger c_x\right) + \lambda\left(c_0^\dagger c_0 + c_1^\dagger c_1\right),
$$

with $\lambda$ the strength of the defect. We can easily derive the reflection and transmission coefficients

$$
r(k) = \frac{\lambda e^{ik}(\lambda - 2\cos k)}{2i\sin k + \lambda(2 - \lambda e^{ik})}
$$
$$
t(k) = \frac{2i\sin k}{2i\sin k + \lambda(2 - \lambda e^{ik})},
$$

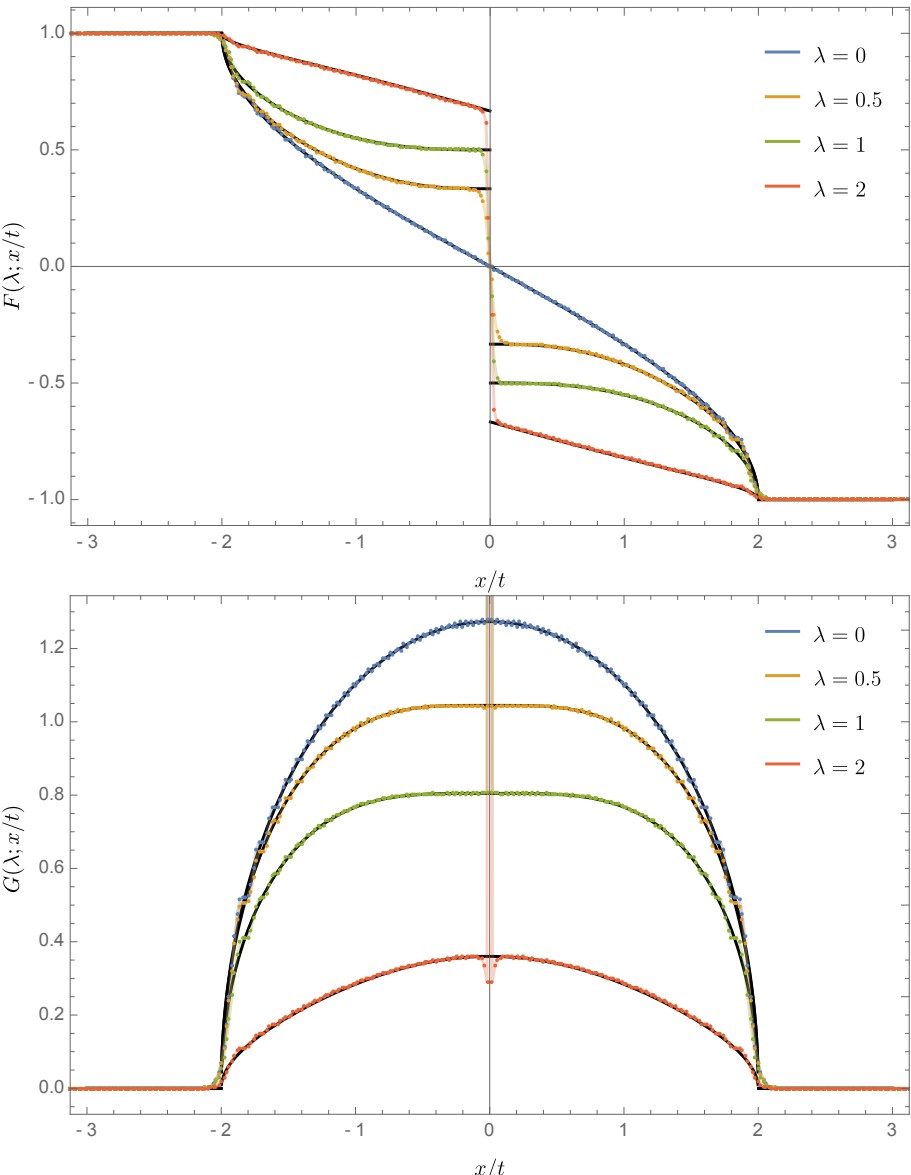

Figure 8: Asymptotics of the density $n_\infty(x/t)$ (*top*) and current $J_\infty(x/t)$ (*bottom*) for a density defect as given by the corresponding scaling functions $F(\lambda; x/t) = -(n_\infty(x/t) - \nu)/\mu$ and $G(\lambda; x/t) = -J_\infty(x/t)/\mu$ respectively for different values of the defect strength $\lambda \geq 0$. Coloured lines and points: numerical data at the maximum time reached ($t = 50$), black lines: analytical expressions from (57) and (58) with $R(k)$ given by (61).

resulting in a reflection probability

$$R(k) = \frac{\lambda^2(\lambda - 2\cos k)^2}{2 + 2\lambda^2 + \lambda^4 - 4\lambda^3 \cos k + 2(\lambda^2 - 1)\cos 2k}. \tag{61}$$

Fig. 8 shows the comparison between analytical and numerical profiles for various values of the defect strength $\lambda$. Only positive values are used since the asymptotic profiles for values of $\lambda$ with opposite signs turn out to be identical. This is due to the symmetry of $R(k)$ in (61) under the transformation $(\lambda, k) \rightarrow (-\lambda, \pi - k)$. The agreement between analytics and numerics is once again perfect everywhere except close to the origin, where as expected finite distance effects due to the defect cause deviations of finite time numerical results from the asymptotic

values.

As a last note, we should comment on the applicability of our analytical method for general inhomogeneous initial states. Even though we have focused our analysis on initial states of the product-state form with a sharp density step between the two adjacent middle sites as described by (19), our method applies to any other initial state that is characterised by different asymptotics on the left and right side far from the middle (independently of its behaviour in the middle) and that is Gaussian with respect to the fermionic fields. This class of states includes not only product-states with smooth density profiles $n_0(x)$, but also non-product states described by two-point correlations $C_0(x, y)$ such that $\lim_{r \to \infty} C_0(x - r, y - r) = C_{0L}(x - y)$ and $\lim_{r \to \infty} C_0(x + r, y + r) = C_{0R}(x - y)$ with $C_{0L}(x) \neq C_{0R}(x)$. To see this, one has to look at our derivation in more detail: What we essentially showed is that the large time and distance asymptotics of correlations is completely determined by the location and residue of the momentum-space pole of the correlation function (37) at equal momenta. The origin of this pole can be traced back to the step-like form of the initial state and its residue can be shown to be determined by only the large distance asymptotics of the initial state correlations and not by the slope of the step or any other intermediate spatial features. The only other relevant assumption in our method is that the momentum-space expression for the correlation function is free from singularities on the real momentum axis, which is guaranteed at least for initial states with exponentially decaying correlations.

On the other hand, typical examples of inhomogeneous initial states in spin chains may not necessarily fall in the category of Gaussian fermionic states (due to the non-linearity of the Jordan-Wigner transformation from spin to fermion variables). For non-Gaussian initial states, the two-point correlation function is no longer sufficient to provide a complete description of the state. However even in this more general case the large time asymptotics under a free fermion Hamiltonian is expected to be described by a Gaussian steady state that keeps only memory of the initial two-point correlations (provided that initial correlations satisfy the clustering property [63, 64]), therefore the same arguments apply.

# 6 Conclusion

In this work we studied the effects of a non-interacting defect on quantum transport after inhomogeneous quantum quenches. We showed in an exact and rigorous way that the asymptotics of particle density and current are correctly given by a simple semiclassical approach, except for finite distance corrections close to the defect. Moreover, we derived exact expressions for these finite distance effects that cannot be captured by the semiclassical approach. Knowledge of their general characteristics (time oscillations and exponential decay with distance) helps to distinguish them from the physically more interesting behaviour of the large distance asymptotics.

An obvious open question that arises is how to handle genuinely interacting defects and whether the previous semiclassical approach can be suitably generalised to describe the asymptotics in this more general case. Our analysis of the non-interacting case suggests that one prerequisite for the exact solution of the interacting problem is to understand the scattering effects induced by the defect on multi-particle states. We leave this exciting problem for future work.

Lastly, it is worth emphasising the analogy between an inhomogeneous quench and the Landauer problem. In both problems the physical interpretation and the mathematical treatment are similar. This analogy points to a broader connection between out-of-equilibrium physics of open and closed extended systems that would be interesting to formulate more rigorously and possibly exploit in order to study more complicated problems in both contexts.

## Acknowledgements

We thank M. Fagotti and B. Bertini for useful discussions.

**Funding information** The work is supported by Advanced grant of European Research Council (ERC) 694544 – OMNES, as well as the grants N1-0025 and N1-0055 of Slovenian Research Agency.

## A    Finite distance effects for $j > 1$

In this appendix, we provide more details on the derivation of finite distance features of the NESS current and density for $j > 1$. These are partially (but not exclusively) due to the presence of bound state eigenstates that are localised at the defect.

We start with the correlation function. The last expression in (32) is still valid, but the eigenstate sum in the propagator (33) should now include also the two bound states. In the thermodynamic limit and large time limit, this modification gives an additional term in (38)

$$
\begin{aligned}
&2\mu \sum_\sigma \phi_{b,\sigma}(x)\phi_{b,-\sigma}(x') \sum_{y=1}^{L/2} \phi_{b,\sigma}(y)\phi_{b,-\sigma}(y) e^{i(E_{b,\sigma}-E_{b,-\sigma})t} \\
&= -\mu \frac{N_b^4}{\cosh\kappa} e^{-\kappa(x+x'-1)} \left[(-1)^{x'} e^{i(E_{b,+}-E_{b,-})t} + (-1)^x e^{-i(E_{b,+}-E_{b,-})t}\right] \\
&= -\mu \frac{\sinh^2\kappa}{\cosh\kappa} e^{-\kappa(x+x'-1)} \left[(-1)^{x'} e^{2iE_{b,+}t} + (-1)^x e^{-2iE_{b,+}t}\right] \\
&= -\mu \frac{(1+j)^2(1-j)^2}{2j(1+j^2)} j^{-(x+x'-1)} \left[(-1)^{x'} e^{2i(j+1/j)t} + (-1)^x e^{-2i(j+1/j)t}\right],
\end{aligned}
$$

where we used (29), (30) and (31) that describe the bound states. Note that there are no contributions from cross terms between bound states and the continuous part of momentum values in the double eigenstate sum of (32). Such cross terms are off-diagonal and therefore decay with time. In contrast the above contribution is diagonal and survives at large times, because both bound states have the same energy.

The above term contributes an additive correction to the large time asymptotics of the current $J(x,t)$

$$
\delta J(x,t) = -2\mu \frac{(1+j)^2(1-j)^2}{j(1+j^2)} (-1)^x j^{-|2x-1|-1} \sin 2(j+1/j)t, \tag{62}
$$

which is reported in the main text in (40).

For the large time asymptotics of the density $n(x,t)$ for $j > 1$, apart from the contribution of the bound states, there is one more difference in comparison with the case $j < 1$: the location of the poles in the integrand of (41) changes so that they do contribute to the integral. Taking into account both of these changes, we finally find

$$
\delta n(x,t) = \mu j^{-|2x-1|-1} \left[ \text{sign}(x-1/2)(1-j^2) + \frac{(1+j)^2(1-j)^2}{(1+j^2)} (-1)^x \cos 2(j+1/j)t \right], \tag{63}
$$

reported in the main text in (43).

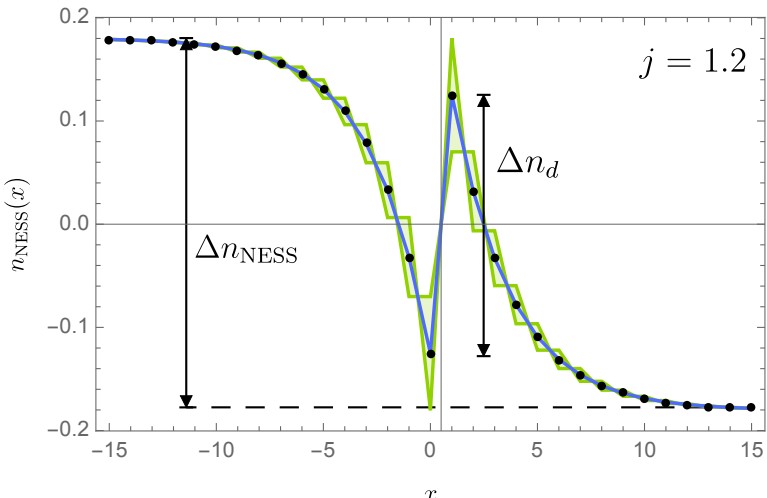

Figure 9: Finite distance features of the NESS density as a function of the position $n_{t\to\infty}(x, t)$ for defect strength $j = 1.2 > 1$ (Eq. (43)). Dots and blue line: time averaged density profile, green lines: amplitude of persistent oscillations about the average value. The average density exhibits a step $\Delta n_d$ (Eq. (45)) between the two defect sites 0 and 1 that is inverted in comparison with the asymptotic averaged difference $\Delta n_{\text{NESS}}$ (Eq. (44)) far from the defect. In contrast, for $j < 1$ the jump between the defect sites is exactly the same as that far from the defect and the density profile is independent of the position, apart from this jump.

# B   Complex analysis tricks

## B.1   Conversion of sums into integrals in the presence of a pole singularity

We consider a sum of the form

$$\sum_{\substack{k=a \\ \text{step: } \delta k}}^{b} f(k),$$

where $f(k)$ is regular at all points $k_n = a + n\delta k$, $n = 0, 1, 2, \dots$ but may have singularities at other points in the interval $(a, b)$ and we want to write it as an integral in the limit $\delta k \to 0$.

We first write the sum as a sum of contour integrals on infinitesimal circles encircling each of the numbers $k_n$ with a suitable integrand $1/(e^{2\pi i(k-a)/\delta k} - 1)/\delta k$ that has at those points first order poles all with residue $1/2\pi i$. Then we merge the contours into a single contour enclosing the straight line interval in which all numbers $k$ lie (i.e. $[a, b]$), but making sure to remove any poles of the function $f(k)$ in this line interval by subtracting their residues. Then we take the $\delta k \to 0$ limit, in which one of the two lines of the contour, the one below, gives a vanishing contribution since $1/(e^{2\pi i(k-a-i\epsilon)/\delta k} - 1) \to 0$. The one above instead gives simply an integral along the real line interval $[a, b]$ shifted by an infinitesimal imaginary number $+i\epsilon$, since $1/(e^{2\pi i(k-a+i\epsilon)/\delta k} - 1) \to -1$.

We therefore find that the original sum converges to an integral along a contour just below

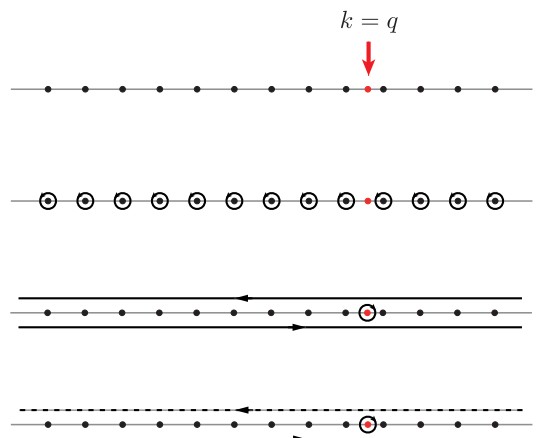

Figure 10: Complex analysis method for the conversion of a sum into an integral in the case of singular integrand.

the real axis and the sum of residues of the corresponding poles (Fig. 10)

$$
\begin{aligned}
\sum_{\substack{k=a \\ \text{step: } \delta k}}^{b} f(k) &= \sum_{\substack{k=a \\ \text{step: } \delta k}}^{b} \oint_{C(k_n, \epsilon)} dk \, \frac{1/\delta k}{e^{2\pi i (k-a)/\delta k} - 1} f(k) \\
&= \left( \int_{a-i\epsilon}^{b-i\epsilon} - \int_{a+i\epsilon}^{b+i\epsilon} \right) dk \, \frac{1/\delta k}{e^{2\pi i (k-a)/\delta k} - 1} f(k) - 2\pi i/\delta k \sum_{k_j: \text{ poles of } f} \frac{1}{e^{2\pi i (k_j - a)/\delta k} - 1} \mathop{\mathrm{Res}}_{k=k_j} f(k) \\
&\xrightarrow{\delta k \to 0} \int_{a}^{b} \frac{dk}{\delta k} f(k - i\epsilon) - 2\pi i \sum_{k_j: \text{ poles of } f} \left( \lim_{\delta k \to 0} \frac{1/\delta k}{e^{2\pi i (k_j - a)/\delta k} - 1} \right) \mathop{\mathrm{Res}}_{k=k_j} f(k).
\end{aligned}
\tag{64}
$$

## B.2 Contour deformation for the calculation of large distance and time asymptotics

We consider an integral of the form

$$
\int_{a-i\epsilon}^{b-i\epsilon} dk \, g(k) e^{+ikx - iE(k)t},
$$

where the function $g(k)$ is assumed to be analytical for real $k$ in the interval $[a, b]$ except at one point $k = q$ where it has a pole, the function $E(k)$ is analytical in the same interval, and $\epsilon$ is an infinitesimal shift of the integration contour, let us say, below the real axis. We want to find its asymptotic value as both $x$ and $t$ tend to infinity, while their ratio $\xi = x/t$ is kept fixed.

To this end, we just have to make sure that the integration contour runs always within the region where the real part of the exponent of $e^{+ikx - iE(k)t}$ is negative, so that the integral decays to zero as $t \to \infty$. The sign of the real part depends on the value of $\xi$ and can be positive along some pieces of the original contour, in which case we have to deform the contour from below to above the real axis along those pieces. If the pole at $k = q$ is crossed while the contour is being deformed, then we have to add its residue which therefore results in a non-vanishing value in the considered limit.

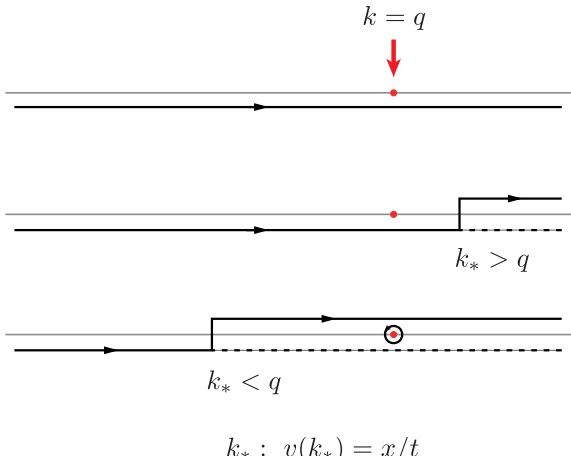

Figure 11: Complex analysis method for the derivation of the asymptotics of the integral.

The condition on which side of the real axis the contour should be shifted at some value of $k$ is $\mathrm{Re}\left[\mathrm{i}((k-\mathrm{i}\epsilon)\xi - E(k-\mathrm{i}\epsilon))\right] < 0$, i.e.

$$\left(\xi - E'(k)\right)\epsilon < 0.$$

Therefore the points at which the shift of the contour should change sign are given by the values $k_*$ that are solutions of the equation

$$E'(k_*) = \xi.$$

If $\xi > E'(q)$ then the pole at $k = q$ is inside a region where the direction of the deformation should be switched to ensure exponential decay of the integral and therefore the pole does contribute, otherwise not. We finally find that

$$\int_{a-\mathrm{i}\epsilon}^{b-\mathrm{i}\epsilon} \mathrm{d}k\, g(k)\mathrm{e}^{+\mathrm{i}(k\xi - E(k))t} \xrightarrow[x/t:\,\mathrm{fixed}]{t\to\infty} 2\pi\mathrm{i}\,\mathrm{e}^{+\mathrm{i}(q\xi - E(q))t}\,\Theta(\xi - E'(q))\mathop{\mathrm{Res}}_{k=q} g(k). \tag{65}$$

Assuming, for example, that $E'(k)$ is a monotonously decreasing function, there is only one solution $k_*(\xi)$ and the pole contributes only when $q > k_*(\xi)$ (Fig. 11).

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
