# Peer review of "Non-equilibrium quantum transport in presence of a defect: the non-interacting case"

_SciPost Physics, doi:SciPost Phys. 6, 004 (2019)_

## Round 2 · Referee Report · Anonymous (Referee 1) · 2018-5-11

Strengths

Works in detail through a simple example

Weaknesses

  1. Not clear what the new results are
  2. Some of the results appear to be very counterintuitive on physical grounds -- which either means they are wrong, or they need a full explanation.

Report

This paper looks at non-equilibrium quantum transport after a quench in the presence of a localised scatterer in a non-interacting system. Most of the paper appears to concentrate on observables in the vicinity of the impurity at large times (the steady state limit), although there are some results also for the propagation of the front following the quench.

While I would agree with the authors that it is sometimes worthwhile to work through a simple system in a lot of detail in order to demonstrate the effectiveness of different methods, or sometimes subtleties that may be overlooked, I find it hard to see the purpose of this paper. The abstract appears to concentrate on the fact that a defect obstructs particle transport (commonly known as resistance), which has been known for 100s of years; and the details of the steady state for this in a non-interacting Fermi system have also been known since the work of Landauer [J. Res. Dev. 1. 233 (1957)] and Buttiker [Phys. Rev. Left. 57, 1761 (1986)], and has been the subject of many reviews and books [see e.g. “Electronic Transport in Mesoscopic Systems” by S. Datta (1995)].

Even the extension of this (in general terms) to the interacting case is known [Meir and Wingreen, PRL, 68, 2512 (1992)], although as the authors rightly point out, there is much work going on at present of different methods to evaluate this in different systems.

While the quench in order to get the population imbalance is sometimes implicit in the formalisms developed in the works above, there has also been a lot of work that explicitly looks at this — see e.g. Branschadel, Schneider and Schmitteckert, “Conductance of correlated systems: real-time dynamics in finite systems”, arXiv1004.4178 (it is published, I just couldn’t find the correct reference), or many other works by Schmitteckert or Heidrich-Meisner, who usually use slightly different quench protocols, which turn out not to be important for the steady state.

In addition to finding it hard to understand what the main point of the paper is and what is new; I believe there are some errors in the results. Concentrating on Eq. 32 (or its equivalent generic version in the unnumbered equation above Eq. 50), this formula is almost the Landauer result except that the integral is over half the Brioullin zone, rather than only over energy levels where there is an imbalance. In addition to the Landauer-Buttiker equation being extremely well checked and verified over the last 50 years, I find Eq. 32 very hard to believe for two reasons: 1. It implies a conductance that is independent of voltage (and of Fermi level), even though the transmission depends on the energy; and 2. It implies that states from from the Fermi surface matter, even for arbitrarily small voltage. My suspicion is that the numerics only agrees with this analytic formula because it was only done for the parameters nu=0, mu=2, meaning that one lead begins completely empty and the other begins completely full — in which case Eq. 32 matches the Landuer-Buttiker result.

With regards to the numerics, it would also be helpful if the authors could explain why they used TD-DMRG, which works in an interacting basis and as such as rather inefficient for non-interacting systems.

In summary, I’m not sure I understand the purpose of this manuscript in the scientific literature, as it appears to be trying to derive something already known for a long time. In addition, I am not convinced that the results presented in the paper are correct — and if they are, the authors need to discuss a) why they disagree with the well-known Landauer-Buttiker approach; and b) answer the two questions above, as the result seem counterintuitive on physical grounds. I therefore cannot recommend this paper for publication.

Requested changes

Major revision needed of almost all points

  • validity: poor
  • significance: poor
  • originality: low
  • clarity: poor
  • formatting: good
  • grammar: good

Author:  Spyros Sotiriadis  on 2018-06-08  [id 268]

(in reply to Report 1 on 2018-05-11)
Category:
remark
reply to objection

The main new results of our work are the exact formulas (39),(41) (and the more general (50),(51)) for the asymptotic density and current profiles after this type of quench, which we derive using the semiclassical approach and later prove rigorously. The profiles of these quantities are of central interest in the analysis of inhomogeneous quantum quenches and to the best of our knowledge none of these formulas has appeared in earlier literature. The referee’s criticism is based on the misconception that our quench problem is equivalent to the conduction problem of Landauer-Büttiker theory and on an out-of-context comparison with the Landauer-Büttiker formula. Moreover he/she has completely overlooked our main findings mentioned above focusing only on our results about the asymptotics at the defect eq.(32).

Landauer-Büttiker theory refers to transport through a system coupled to infinite external baths at thermal equilibrium, which is different from the quench in terms of physical settings and questions: In the quench problem the system is extended and closed (the dynamics is unitary) and we are interested in the spatial profile of observables which are time-dependent for all times. In the Landauer-Büttiker problem the system is open, the internal dynamics of the baths is ignored (they are assumed to have fixed thermal spectral densities) and we are interested in the (time-independent) conductance as a function of an external voltage (Fermi level difference). Therefore a rigorous analytical solution of the inhomogeneous quench problem cannot be drawn from Landauer-Büttiker theory. However, as we already discuss in the conclusions, there is a physical analogy between the two problems as far as only the NESS at the defect is concerned. The Landauer-Büttiker formula is compatible with our semiclassical solution in the special case of thermal momentum distributions on the left and right half of the system.

As for the apparent contradiction of our eq.(32) with the Landauer-Büttiker formula, the referee seems to have confused the quench initial state with low-temperature thermal states with Fermi level difference given by the left/right density difference $\mu$, which is incorrect. If one tries to interpret the two halves of the system as baths in contact with the defect in analogy with the Landauer-Büttiker settings, then the effective bath spectral densities corresponding to the inhomogeneous quench initial state are generally non-thermal: they are rather described by Generalised Gibbs Ensembles corresponding to the homogeneous quenches on the left and right side asymptotically far from the origin. This means that the left/right density imbalance is generally non-zero in all energy levels (as typical in quench problems), in contrast to the low-temperature thermal states assumed in the Landauer-Büttiker case. For the same reason the density imbalance cannot be generally described in terms of a voltage, i.e. difference between Fermi levels of thermal distributions. In the numerics supporting our formulas we used an initial state with momentum densities given by eq.(2) and values $\nu=0$ and $\mu=0.1$. Based on the above, we don’t think that any explanation is needed since eq.(32) should not be equal to the Landauer-Büttiker formula. We do however plan to add some comments showing how the Landauer-Büttiker formula can be seen as a special limit of our semiclassical formula (unnumbered eq. after (9)) in the case of low-temperature thermal initial states.

---

## Round 2 · Referee Report · Anonymous (Referee 2) · 2018-6-1

Strengths

(1) Numerous methods used (exact analytics, semi-classical hydrodynamics, numerical)
(2) Problem is well-defined
(3) Main result is compact and distinct from what happens in clean free-fermion systems: the density discontinuity survives at long times

Weaknesses

(1) Not clear why numerical approach is needed for this problem. The problem is analytically solvable, and one may numerically diagonalize the Hamiltonian (and simulate the entire quench setup) using a single $N\times N$ matrix.

(2) Perhaps I'm misunderstanding, but Figure 8 appears to show a fermion density which ranges from -1 to 1. I'm used to mapping magnetization in the range (-1/2,1/2) to fermion density which varies from 0 to 1. This could be a simple rescaling issue, but I do not know how to interpret $\left\langle c^{\dagger}c\right\rangle < 0$.

Report

The authors consider a one-dimensional system of free fermions which are arranged in a domain wall density profile at t = 0 and allowed to evolve in time. As noted, the clean problem has been considered in numerous contexts with various interactions or other modifications to the system. In this work, the authors consider a defect at the center of the system so that transport is impeded. Using the semi-classical hydrodynamic approach, exact analytics and tDMRG, the long-time value of the density jump across the defect is obtained and found to be non-vanishing. Particle current is also calculated.

It is unclear to me why tDMRG is used at all in this work. The comparison between exact analytics and tDMRG provides some level of support for trusting the tDMRG where analytic results are unavailable, but no such interacting regimes are ever considered in this paper. I feel some sort of interacting calculation should be included to make use of the tDMRG or this numerical approach should just be removed, as it adds nothing to the results that are currently included. For a simple check of the analytic work, it's possible to reproduce all the tDMRG results by simple matrix diagonalization:

For the quench, one can think of the initial state as the ground state of the Hamiltonian

\[H_{i} = -\sum_{x}[c_{x}^{\dagger}c_{x+1} + c_{x+1}^{\dagger}c_{x}] - h\sum_{x}(x-\frac{N}{2})c_{x}^{\dagger}c_{x}\]

In the limit $h\rightarrow \infty$, the domain wall state is obtained. The time evolution takes place under $H_{f}$ which is given in the text as the unlabeled formula between Eq. (12) and Eq. (13). These may be formally diagonalized

\[H_{i} = \sum_{l}\lambda_{l}\gamma_{l}^{\dagger}\gamma_{l}\]
\[H_{f} = \sum_{m}\epsilon_{m}\eta_{m}^{\dagger}\eta_{m}\]

where these new operators are related to $c_{x}$ by linear transformations

\[\eta_{m} = \sum_{n}V_{nm}c_{n}, \;\;\;\;\;\;\;\;\;\; \gamma_{l} = \sum_{n}U_{nl}c_{n}\]

Performing the diagonalization numerically, the $U$ and $V$ are matrices containing the eigenvectors of $H_{i}$ and $H_{f}$, and all observables can be obtained in terms of these matrices and the $\epsilon_{m}$. Such a calculation is much less involved than using tDMRG.

I'm not aware of other calculations using the free fermion chain with a central defect in exactly this setup (though there are examples of other calculations in similar systems, such as entanglement entropy V. Eisler and I. Peschel, EPL 99, 20001 (2012)), so the main result is publishable. The hydrodynamic approach is very intuitive and provides a simple picture for the dynamics.

In summary, I think the core of the work is a publishable result, but the numerical approach seems rather unsuited for this kind of calculation. The density discontinuity which persists is completely different from what happens in other clean systems, both interacting (L. Piroli et al, Phys. Rev. B 96, 115124 (2017)) and non-interacting (J. Lancaster, Phys. Rev. E 93, 052136 (2016)), where this discontinuity always seems to be smoothed out over time. Either using the tDMRG for an interacting case or placing the analytic work in the context of other domain-wall investigations with fermions systems could accomplish this.

Requested changes

(1) In Introduction, "Energy or particle current" is somewhat vague given that these are two distinct quantities (i.e., Ref [5]). I understand that either could be computed, but it appears only particle current is actually calculated in this paper.

(2) Figure 1 is missing a color density scale. Sometimes it is hard to extract quantitative information from color maps, so either a density scale or another actual snapshot plot of m(x,t) vs x at some fixed time could help (referencing Figure 8, if these correspond to the same scenario would make me shut up about this).

(3) Rework paper to use tDMRG for what it's really helpful doing or remove the numerical treatment and flesh the main result.

(4) Some vague/confusing wording should be fixed. Example:

  • bottom of p. 25 "Our analysis of the non-interacting case suggests that solving the general problem passes through understanding the scattering effects induced by the defect on multi-particle states" ("passes through understanding" is unclear phrase)

  • validity: good
  • significance: low
  • originality: good
  • clarity: low
  • formatting: excellent
  • grammar: excellent

Author:  Spyros Sotiriadis  on 2018-06-08  [id 269]

(in reply to Report 2 on 2018-06-01)
Category:
remark

We would like to thank the referee for the careful reading of the manuscript and constructive comments.

The use of tDMRG numerical method is certainly a heavy tool for the study of a non-interacting problem. The referees are right that numerical diagonalisation is in principle more suitable and efficient in the present setup. In fact this choice of numerical method was incidental simply because we used our existing tDMRG code to produce data for more general types of defect including interactions and this is a special set of those data. However we have double-checked their accuracy using exact diagonalisation for some parameter values. We could remove the numerics since the exact analytical calculations are the main point of this work, however we think that certain aspects like the details of the front profile that we didn’t derive analytically are still worth to display. We will clarify in the text that exact diagonalisation is more suitable for the numerics.

The referee is also right about the rescaling issue in fig.8. We will fix that together with the other changes requested in the next version.

---

## Round 2 · Referee Report · Anonymous (Referee 3) · 2018-6-27

Strengths

1- The problem is timely and interesting 2- The paper is accessible also to researchers who are not experts in the field 3- The results are obtained with different methods and are checked against numerics

Weaknesses

1- The goal of the paper is unclear

Report

This paper investigates the transport properties of an out-of-equilibrium quantum many-body system in the presence of a localised defect. In particular, the authors study the time evolution of charges and currents in a simple state that is let to evolve under a free-fermion Hamiltonian with a generic localised defect preserving the $U(1)$ symmetry.
The main results are obtained by extracting the asymptotic behaviour of the exact time evolution in the limit of large time. In addition, the authors show that the leading behaviour is captured by a simple semiclassical approach.
As far as I can see, both the results and the methods are correct, and I think that the authors have correctly answered some of the criticisms raised by the previous referees. There is however a criticism that, in my opinion, is still alive: the goal of the paper is unclear. The semiclassical approach considered in this paper is a simple application of what is already established, so Eqs (50) and (51) are not a sufficient motivation for a paper. On the other hand, the analytic proof of the semiclassical result is very interesting, indeed the semiclassical picture, despite being simple and reasonably correct, is not a priori valid. For this reason, I'm wondering whether this paper could be more appropriate to a more mathematical journal like Journal of Physics A. In any case, the authors should clarify the goal of the paper: there are already several works on this or on similar topics (e.g., [22], [23], [24], [25], and [35]]), and the generic motivation invoked by the authors ("In order to better understand the physics of quantum transport after inhomogeneous quenches in the presence of defects, it is instructive to first study the case of non-interacting defects, which can be analysed analytically") is not enough anymore. If the authors address this issue, the paper could become suitable for publication in scipost; in the present form, I think that the paper is more appropriate to a different journal.

Requested changes

  1. The goal of the paper should be stated very clearly in the introduction, in such a way that a reader could clearly understand the difference between this work and the previous ones.
  2. At the beginning of page 3, Refs [7,8] are cited as the papers were it was shown that a different ensemble can be associated with the asymptotics at each ray of fixed distance/time ratio. As far as I know, this was first done in [25], which also investigated dynamics much closer to the ones investigated in this paper.
  3. The approach described below eq. (2) is essentially the same considered in [Bernard, Doyon, and Viti, J. Phys. A 48 (2015) 05FT01] and in [22].

  • validity: good
  • significance: good
  • originality: ok
  • clarity: high
  • formatting: excellent
  • grammar: excellent

Author:  Spyros Sotiriadis  on 2018-07-10  [id 289]

(in reply to Report 3 on 2018-06-27)
Category:
answer to question

We would like to thank the referee for his comments, which we will take into account in the next version of our paper. Regarding the requested changes:

  1. The main difference between our work and other related works on inhomogeneous quenches in non-interacting systems is that, as the referee recognises, we derive the exact asymptotics from first principles instead of using the semiclassical approach without proving that it is exact. Our mathematical treatment of this problem may serve as a preliminary step towards the exact calculation of the asymptotics in the interacting case. On the other hand, we show that deviations from the semiclassical results can appear close to the defect. We will clarify these differences both in the abstract and in the introduction in the next version. We don't see why the mathematical-physics character of our work makes it unsuitable for SciPost, as there don't seem to be such content restrictions. In fact this manuscript has been submitted indicating math-phys as a secondary category and we wouldn't mind setting it as primary. We let the editors judge about this.

  2. Indeed ref.[25] is the right reference to be cited at that point. We will fix that in the next version.

  3. To be precise, the semiclassical approach we discuss here is different from the approach of those references: In both [Bernard, Doyon, and Viti, J. Phys. A 48 (2015) 05FT01] and [22] the analysis is based on operator algebra, i.e. on the exact time evolution of operator-valued fields under a CFT with a defect, while in that section we present only a semiclassical approach based on the physical picture of well-separated quasiparticles moving ballistically. Even though it’s certainly true that conceptually these works are close to ours and we will include them in our references, it would perhaps devalue those works to cite them at that point of the semiclassical section. Our argument there is much closer to the one presented in the earlier work [32], even though that one discusses a problem with no defect present.

---

## Round 3 · Referee Report · Anonymous (Referee 4) · 2018-11-24

Strengths

  1. Analysis is carried out by detailed first principle computations.
  2. The paper provides the unifying framework to study transport in non-interacting impurity systems.
  3. Exact asymptotics of the NESS current and density are obtained.

Weaknesses

The methods developed in the paper are not applicable to interacting impurity models.

Report

I do acknowledge the weaknesses pointed out by other referees in the first version, but I think the authors managed to overcome these criticisms by addressing the comments raised. The goal of this paper, the exact determination of the asymptotics of the NESS current and density, is now clearly stated in the introduction. I therefore recommend for the publication in SciPost as it is.

Requested changes

  1. I think eq. (24) is not valid at $x=0$, only for $x>0$.

  • validity: high
  • significance: good
  • originality: good
  • clarity: high
  • formatting: excellent
  • grammar: good

Author:  Spyros Sotiriadis  on 2018-12-04  [id 364]

(in reply to Report 1 on 2018-11-24)

We would like to thank the referee for his comments.

Indeed eq.(24) as introduced (general solution of the recursive equation in the right half side) holds only for x>0 not for x=0. We have corrected this in eq.(24) and everywhere else in the paper. Note that this doesn't affect the analytical derivation of the asymptotics at the middle because the latter was based on the reflection symmetric expressions in full space.

---

## Round 3 · Referee Report · Anonymous (Referee 2) · 2018-11-25

Strengths

  1. Highly-detailed discussion of a particular physical situation using analytic approximations and numerical methods.

  2. Validation of semi-classical approximation in highly non-equilibrium setting with inhomogeneous initial state-an interesting result.

  3. Authors have improved the manuscript based on the feedback of previous referee reports

Weaknesses

  1. Some unclear statements remain

Report

The authors consider non-interacting tight-binding dynamics in a system of fermions with a centralized defect which separates two semi-infinite regions of constant particle density. Using semi-classical arguments, the particle density and current are calculated at long times. Numerical results are obtained using tDMRG, ultimately showing that the simple analytic approach captures the long-time limit of simple observables quite well.

The comments from my previous review were taken into full consideration, and I find that the authors have also made revisions in accordance with the suggestions put forth by other reviewers. At the level of my own understanding of the criticisms raised by other reviewers, I am satisfied with the responses and changes made in the revised manuscript.

I have only a few minor suggested changes which are included below. These remaining points mainly concern the wording of certain statements and some seemingly simple generalizations of the the results presented. In particular, Ref. [5] includes results for domain walls which are of arbitrary "size," while the authors only consider a maximal jump from a fermion density of 0 to density 1. This is understandable in the tight-binding language (as opposed to the equivalent "spin language") because the linearly-varying chemical potential naturally leads to this maximal state, regardless of the slope. However, I am curious as to whether generalization of the present results are straightforward or hindered by the complicated nature of the generalized domain wall state presented in Ref. [5].

Requested changes

  1. The phrase "It has been shown that an inhomogeneous quench starting from a step profile allows for an exact solution" appears in the introduction. I'd argue that any type of chemical potential allows for an "exact" solution in the sense that the problem can be formally diagonalized. A linearly-varying potential can be exactly diagonalized for any slope (T. Hartmann et al, New J. Phys. 6, 2 (2004), and many others). But I would agree with the more careful statement that the expressions are cleaner and perhaps more useful for the sharp step profile (caused by an infinite field gradient).

  2. In the introduction, the phrase "KAM" is used without explanation. It's likely clear to many that this refers to the Komolgorov-Arnold-Moser theorem, but I think the acronym should be introduced.

  3. The conclusion contained the statement “Lastly, it is worth to stress the analogy between an inhomogeneous quench and the Landauer problem....” I feel this should should read something like “lastly, it is worth emphasizing the analogy between….” (minor point of grammar)

  4. If tDMRG is being used to treat a truly non-interacting problem, I would like to see that the results are as general as possible. Concerning the last paragraph in my report, is it possible to consider more general domain wall states (as considered in Ref. [5]) in this framework using your tool kit, or does something make this technically difficult?

  • validity: high
  • significance: good
  • originality: high
  • clarity: good
  • formatting: perfect
  • grammar: excellent

Author:  Spyros Sotiriadis  on 2018-12-04  [id 363]

(in reply to Report 2 on 2018-11-25)

We would like to thank the referee for his comments, which we have taken into account in the new version. In more detail:

  1. We agree that the full dynamics resulting from any initial density profile in this free system can be solved formally by exact diagonalisation. What we meant instead was "exact derivation of the asymptotics in closed-form expressions". In fact we should also clarify that closed-form expressions are possible also for smooth steps and some of the references cited at that point refer indeed to smooth profiles. We have changed the sentence accordingly.

  2. Indeed, KAM should have been written explicitly as Kolmogorov-Arnold-Moser. We have replaced the acronym.

  3. We have corrected this sentence.

  4. This is an interesting point: Even though we focused our analytical proof on initial states corresponding to a density step with values $\mu\pm\nu$ changing sharply between the two middle sites 0 and 1, our method applies to any other initial state that is Gaussian and characterised by different asymptotics far on the left and right side, independently of the behaviour in the middle. This class of states includes the partitioning protocol of ref.[5]. We have added two paragraphs at the end of section 5 to comment on this. The main idea is that the asymptotic formulas are derived from the “kinematical” pole at equal momenta of the expression for the correlation function after a quench, which in turn is determined only by the large distance asymptotics of the initial state: e.g. if we consider a density profile given by a Heaviside step function or by $\tanh(ax)$ (or any other smooth profile), then we can see that in both cases the Fourier transform has the same pole $i/k$ as $k\to 0$, independently of the slope parameter $a$ in the smooth case. More details will be given in future work.

---

## Round 3 · Author Response

Following the referees' suggestions, we resubmit a new version of our paper with an improved abstract and introduction that clarify better the main goal of our paper, that is the rigorous derivation of the exact asymptotics after an inhomogeneous quench in presence of a defect, which allows us to demonstrate the validity of the semiclassical approach and identify potential corrections. We have also included a comparison with Landauer's theory and corrected the discussion on numerics as well as several typos and other issues raised by the referees.

---

## Round 3 · List of Changes

1. Improved abstract and extensively modified the introduction (mainly in p.3) to clarify the goal of the paper, also added further information on earlier work including additional citations (on integrability, CFT or Landauer-theory based techniques)
2. added subsection 2.3 on comparison with Landauer’s theory in the special case of thermal baths
3. added discussion on numerical methods, explaining that exact diagonalisation is more efficient for this problem

4. added citations on use of Landauer’s theory to similar problems
5. changed citation for ray dependence of NESS to (Bertini Fagotti, PRL 117(13) 130402 (2016)),
6. added citation [36] (Bernard, Doyon, Viti, J. Phys. A 48 (2015) 05FT01)
7. added colour scale in fig.1
8. fixed rescaling issue in figures 5,6,8
9. corrected mistyped eqs.(55-58) for density/current asymptotics for general defect
10. fixed sign typo in eq.(9)
11. corrected confusing wording in p.25 and elsewhere

---

## Round 4 · Author Response

We would like to thank the referees for their comments. We have modified the manuscript accordingly.

---

## Round 4 · List of Changes

1. Corrected typo in eq.(24).

2. Modified wording in certain sentences following referee 2 comments.

3. Added two paragraphs at the end of sec. 5 with a discussion on the applicability of our analytical method to more general inhomogeneous initial states.

---

## Editorial Decision

published